# IDENTITY-DISENTANGLED ADVERSARIAL AUGMENTATION FOR SELF-SUPERVISED LEARNING

## ABSTRACT

Data augmentation is critical to contrastive self-supervised learning, whose goal is to distinguish a sample's augmentations (positives) from other samples (negatives). However, strong augmentations may change the sample-identity of the positives, while weak augmentation produces easy positives/negatives leading to nearly-zero loss and ineffective learning. In this paper, we study a simple adversarial augmentation method that can modify training data to be hard positives/negatives without distorting the key information about their original identities. In particular, we decompose a sample $x$ to be its variational auto-encoder (VAE) reconstruction $G(x)$ plus the residual $R(x) = x - G(x)$, where $R(x)$ retains most identity-distinctive information due to an information-theoretic interpretation of the VAE objective. We then adversarially perturb $G(x)$ in the VAE's bottleneck space and adds it back to the original $R(x)$ as an augmentation, which is therefore sufficiently challenging for contrastive learning and meanwhile preserves the sample identity intact. We apply this "identity-disentangled adversarial augmentation (IDAA)" to different self-supervised learning methods. On multiple benchmark datasets, IDAA consistently improves both their efficiency and generalization performance. We further show that IDAA learned on a dataset can be transferred to other datasets.

## 1 INTRODUCTION

Empowered by deep neural networks and the computational capability of recent hardware/infrastructure, machine learning (ML) has achieved great breakthroughs on some challenging problems when sufficient labeled data is available. However, deep learning is known to be data-hungry and annotating data in many domains, e.g., medical care or predictions of protein structures, are either difficult or expensive. To overcome this limitation, self-supervised learning (SSL) methods train a model on unlabeled data in a supervised learning manner using self-generated labels by manipulating the data, e.g., rotation degrees (Gidaris et al., 2018), Jigzaw puzzle solutions (Noroozi & Favaro, 2016), clustering (Caron et al., 2018), back-translation (Zhu et al., 2017), etc. These methods recently start to perform on par with supervised learning and exhibit potential to even surpass it Chen et al. (2020); Chen & He (2021). Moreover, their learned representations can be generally applied to different downstream tasks.

Many widely-used SSL methods are built upon sample-identity preservation tasks, e.g., contrastive learning (Oord et al., 2018; Tian et al., 2020a; Chen et al., 2020) and consistency regularization (Chen & He, 2021; Grill et al., 2020; Caron et al., 2020), which aim at learning representations that can preserve the identity of the original sample after applying data augmentations and thus distinguish the augmentations of different samples. For example, contrastive learning targets on an representation space in which a sample (anchor) is closer to its own augmentations (positives) than other samples or their augmentations (negatives). The effectiveness of contrastive learning therefore heavily depends on the quality of data augmentations.

Most SSL methods (Ye et al., 2019; Chen et al., 2020; He et al., 2020; Chen & He, 2021) utilize pre-defined data augmentations to generate positives and negatives. However, as shown in the example (green point) of Fig. 1, they are not adaptive to the data manifold in the embedding space during training and the generated augmentations can be too easy for the sample identification task. In practice, these SSL methods need tens to hundreds times of epochs required by supervised learning to reach comparable performance on the downstream classification tasks (Chen et al., 2020; He et al., 2020; Chen & He, 2021). For the same reason, large batch-size is common and necessary for contrastive learning in order to involve sufficient hard negatives. Therefore, how to modify the positives/negatives for more informative contrastive loss is a critical yet open challenge towards more efficient SSL. Learnable and adaptive data augmentations have been explored for supervised

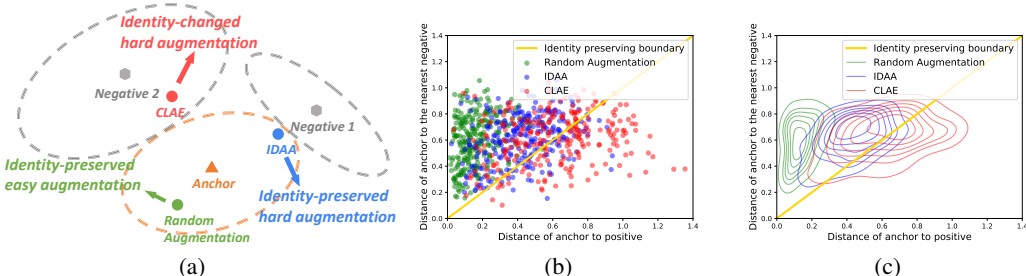

Figure 1: Data augmentations for contrastive Learning: IDAA, CLAE and random augmentation. (a) Identity-preserving and hardness of data augmentations: random augmentation generates identity-preserved but easy samples, CLAE adversarially generates hard but identity-distorted samples, while IDAA (ours) can generate hard and identity-preserved samples. (b) Identity-preserving and hardness of augmentations by different methods: Points below the boundary line have the identity changed, while points close to the boundary are hard samples. (c) Kernel density plot for (b): IDAA generates hard samples without changing the identities.

learning (Cubuk et al., 2018; 2020; Liu et al., 2021) but have not been widely studied in SSL. Probably the most related augmentation method for SSL is CLAE (Ho & Vasconcelos, 2020), which generates hard positives/negatives by adversarially attacking the input. However, as shown in Fig. 1, adversarial augmentations may change the original sample identity and it is infeasible to tune the attack strength for every sample to preserve the identity.

Hence, another challenge for data augmentation in SSL is how to preserve sample identity. Although hard positives/negatives might be achievable by stronger augmentations or adversarial augmentations listed above, they can also distort the true identities of the original samples as the red point in Fig. 1 so the model may erroneously identify other different samples or their augmentations as the anchor sample. Training with those identity-changed augmentations might lead to trivial solutions for SSL and poor representations.

To address the aforementioned two primary challenges of contrastive learning, we have to consider how they interfere with each other. For better efficiency, the data augmentation needs to generate positives and negatives as challenging as possible for the model to distinguish the sample identity, but it should not remove or distort the minimum necessary information retaining the true identity. Thereby, the sample identification task is neither too trivial nor infeasible to learn. In this paper, we study how to automatically generate data augmentations that fulfill the above conditions and improve both the efficiency and effectiveness of current self-supervised learning. We relate the objective of variational antoencoder (VAE) with the sample identification task in contrastive learning from an information-theoretical perspective, which inspires us to disentangle the identity-essential information of an input $x$ as the residual of VAE reconstruction $G(x)$, i.e., $R(x) = x - G(x)$. As illustrated in Fig. 2, in order to modify $x$ to be more challenging in terms of sample identification, we propose to apply adversarial perturbations to the bottleneck features of VAE, which maximize the contrastive loss in an $\epsilon$-ball and results in a modified $G'(x)$. We then utilize $x' = G'(x) + R(x)$ as a data augmentation of $x$ so the identity information captured by $R(x)$ remains intact in $x'$.

Our method, called "identity-disentangled adversarial augmentation (IDAA)", only needs a pre-trained VAE model and does not require any labeled data. In the experiments on multiple benchmarks, when applied to different SSL methods, this simple yet principal data augmentation approach consistently brings improvements on both the efficiency and the performance on downstream tasks. In addition, we present a thorough ablation study to analyze the influence of hyperparameters (e.g., VAE hyperparameters, bottleneck dimensions, $\epsilon$ controlling the attack strength) and experimental settings (e.g., batch size, training epochs, model architecture) on the contrastive learning's performance.

## 2 BACKGROUND

**Contrastive learning** (CL) (Wu et al., 2018; Zhuang et al., 2019; Tian et al., 2020a; Ye et al., 2019; Kalantidis et al., 2020; Hu et al., 2021; Robinson et al., 2020) aims at learning representations that can distinguish different samples and their augmentations. Specifically, it formulates this sample-identification task as a classification problem on each sample (anchor), where the positives are its own augmentations and the negatives are other samples and their augmentations. A widely-used loss for contrastive learning is "InfoNCE" (Tian et al., 2020a) built upon the positives and negatives created by data augmentations. For each mini-batch $\mathbf{x} = \{x_1, x_2, \cdots, x_N\}$ of size $N$, InfoNCE loss

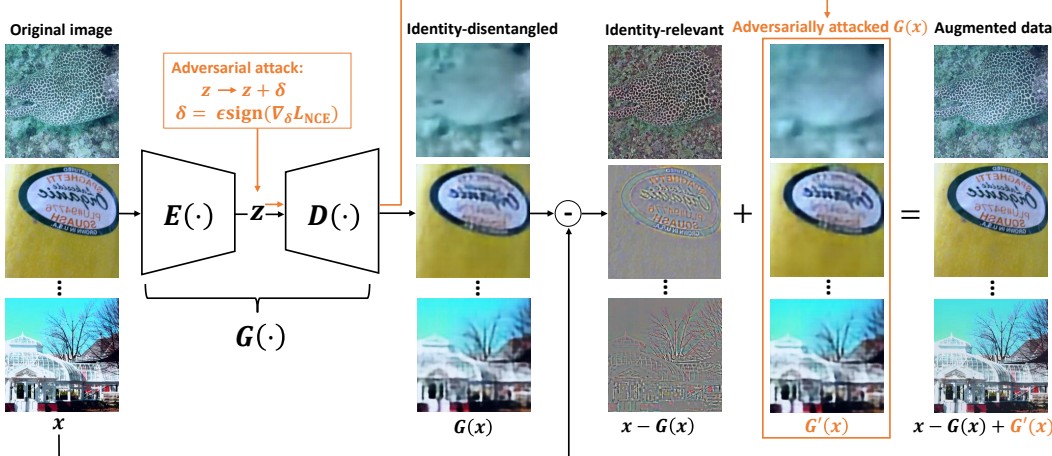

Figure 2: Architecture and pipeline of Identity-Disentangled Adversarial Augmentation (IDAA).

is computed as:

$$L_{\text{NCE}}(\mathbf{x}) = -\frac{1}{N} \sum_{i=1}^{N} \log q_{\text{NCE}}(i|x = x_i), \quad q_{\text{NCE}}(i|x = x_i) \triangleq \frac{\exp \langle f(A(x_i)), h(B(x_i)) \rangle}{\sum_{j=1}^{N} \exp \langle f(A(x_i)), h(B(x_j)) \rangle},$$
(1)

where $\langle \cdot, \cdot \rangle$ denotes inner product, $A(\cdot)$ and $B(\cdot)$ denote two different transformations for data augmentation, $f(\cdot)$ and $h(\cdot)$ are the embedding networks that can be either identical (Chen et al., 2020; Ye et al., 2019) or different (Misra & Maaten, 2020; He et al., 2020), and $q_{\text{NCE}}(i|x = x_i)$ is the softmax probability of identifying the augmentation $B(x_i)$ is transformed from the original sample $x_i$. The choices of data augmentation are critical to the performance and efficiency of CL: weak augmentations might already result in fully distinguishable representations over different samples and nearly zero CL loss, while strong augmentations may overly distort the identity of a sample and make CL too challenging, infeasible, or inefficient. In previous works such as SimCLR (Chen et al., 2020), different (compositions of) augmentations can result in large gaps on CL's performance but finding the best one needs brute-force search of all possible transformations and their compositions, which is usually computationally forbidden in practice.

Another challenge in CL is to keep a large memory bank or mini-batch to cover sufficient amount of "hard negatives" because a sample might be distant from most other samples (and their augmentations) and hard negatives close to it are sparse in the training set. Wu et al. (2018) trains a non-parametric classifier to maximally scatter the representations of all samples over a unit sphere. However, it needs to build a memory bank to store all these representations, which is infeasible for large-scale datasets. He et al. (2020) instead maintain a fixed-sized dynamic dictionary based on a FIFO queue of representations of incoming mini-bathes during training. It finds that a relatively large dictionary is critical to the success of CL. The lack of hard negatives forms a bottleneck to the sample efficiency of CL. Therefore, how to select or modify the data to increase the chance of including more hard negatives is a significant open problem for CL.

**Consistency regularization** consistency regularization (CR) has been studied in several recent works (Chen & He, 2021; Grill et al., 2020; Caron et al., 2020) for self-supervised or semi-supervised learning, which achieves comparable or even better performance than CL in certain scenarios. Compared to CL, CR removes the comparison to negatives and only focus on maximizing the similarity between model outputs for two augmentations of the same sample, e.g.,

$$L_{\text{CS}}(\mathbf{x}) = -\sum_{i=1}^{N} \frac{\langle f(A(x_i)), h(B(x_i)) \rangle}{\|f(A(x_i))\| \cdot \|h(B(x_i))\|},$$
(2)

Similar to CL, CR also aims at preserving the sample identity on the learned representations and it heavily relies on the choice of data augmentations. For example, FixMatch (Sohn et al., 2020) chooses to apply multiple weak augmentation for generating the pseudo-labels (e.g., $f(A(\cdot))$) and strong augmentations to the other branch $h(B(\cdot))$.

# 3 IDENTITY-DISENTANGLED ADVERSARIAL AUGMENTATION

In this section, we will firstly relate the sample-identification task broadly used in designs of self-supervised learning with the training objective of variational auto-encoder (VAE). Specifically, we will show that VAE's training tries to remove the sample-identity related information from its bottleneck features. Hence, the residual of VAE reconstruction may retain the sample-identity information that we wish to preserve in the data augmentation for self-supervised learning. We then propose a data generation and augmentation model based on VAE and analyze the lower bound for identity preservation in its augmented data. In the end of this section, we apply adversarial attack methods to this data augmentation model, which modify samples to be hard positives and negatives for more efficient contrastive learning without changing their sample-identities.

## 3.1 IDENTITY-DISENTANGLEMENT IN CONTRASTIVE LEARNING

The goal of sample-identification can be formulated as maximizing $q(y = i|x = x_i)$, i.e., the likelihood estimation of identifying $x_i$ or its augmentations as sample-$i$, where $x$ and $y$ are two random variables for a sample and an identity label, respectively. The following proposition shows that $q(y = i|x = x_i)$ provides a lower bound for the mutual information $I(x; y)$ between $x$ and $y$.

**Proposition 1.** (Sample-identification likelihood as a lower bound of $I(x; y)$). *If $\mathbf{x}$ is a random mini-batch of size $N$ and the sample-identification likelihood estimation of $x_i$ on its correct identification label $y = i$ to be $q(y = i|x = x_i)$, the mutual information $I(x; y)$ can be lower bounded by*

$$I(x; y) = \log N + \mathbb{E}_{\mathbf{x}} \left[ \frac{1}{N} \sum_{i=1}^{N} \log p(y = i|x = x_i) \right] \geq \log N + \mathbb{E}_{\mathbf{x}} \left[ \frac{1}{N} \sum_{i=1}^{N} \log q(y = i|x = x_i) \right].$$
(3)

where $p(\cdot)$ is the true probability and $q(\cdot)$ denotes an estimation of $p(\cdot)$. A detailed proof is given in the Appendix. In contrastive learning, $p(y = i|x = x_i)$ can be modeled and estimated by $q_{\text{NCE}}(i|x = x_i)$ defined in Eq. (1) using neural network embedding $f(\cdot)$ and $h(\cdot)$ of data augmentations $A(\cdot)$ and $B(\cdot)$. Hence, InfoNCE loss can provides a lower-bound of $I(x; y)$, i.e.,

$$I(x; y) \geq \log N + \mathbb{E}_{\mathbf{x}} \left[ \frac{1}{N} \sum_{i=1}^{N} \log q_{\text{NCE}}(i|x = x_i) \right] = \log N - \mathbb{E}_{\mathbf{x}} \left[ L_{\text{NCE}}(\mathbf{x}) \right].$$
(4)

The above lower bound relates the mutual information $I(x; y)$ and contrastive learning: increasing the batch size $N$ and/or minimizing the InfoNCE loss can improve the tightness of the lower bound and results in representations with better capability on sample identification. Due to the natural sparsity of hard positives and negatives in the original data $\mathbf{x}$, data augmentations and perturbations of $\mathbf{x}$ are necessary to improve the sample efficiency when minimizing $\mathbb{E}_{\mathbf{x}} \left[ L_{\text{NCE}}(\mathbf{x}) \right]$. However, without any constraints, strong data augmentations may remove the identity-related information of some data and change their identities, which results in $q_{\text{NCE}}(i|x = x_i) \ll p(y = i|x = x_i)$, a loose lower bound in Eq. (4), and poor representations via contrastive learning.

## 3.2 IDENTITY-DISENTANGLEMENT VIA VARIATIONAL AUTO-ENCODER

Next, we will show that the training objective of variational anto-encoder (VAE) (Kingma & Welling, 2013) is also related to $I(x, y)$ and sample identification. In particular, VAE aims at removing identity-specific information from the bottleneck features $z$, i.e., minimizing $I(z, y)$. With an output $G(x)$ decoded from $z$, VAE naturally disentangles the most identity-relevant information from $x$. If we can remain such information, i.e., $x - G(x)$, intact in data augmentations and only perturb the rest part $G(x)$, the above problem of contrastive learning can be resolved.

**Lemma 1.** (VAE objective and $I(z; y)$ from Eq. (29) in Appendix B of Alemi et al. (2016)). *Assume that the bottleneck features of VAE are denoted by $z$, the encoder is $E(\cdot)$ and produces distribution $p_E(z|x)$, the decoder is $D(\cdot)$ and produces distribution $q_D(x|z)$, the prior for $z$ is $p(z)$, and the KL-divergence regularization in the VAE objective $L_{\text{VAE}}$ has a weight $\beta$, we have:*

$$- I(z; x) + \beta I(z; y) \leq L_{\text{VAE}},$$
(5)

$$L_{\text{VAE}} \triangleq - \int \mathrm{d}x p(x) \int \mathrm{d}z p_E(z|x) \log q_D(x|z) + \beta \frac{1}{N} \sum_{i=1}^{N} \mathrm{D_{KL}} \left( p_E \left( z|x = x_i \right) || p(z) \right),$$
(6)

In $L_{\text{VAE}}$, $\beta$ (Higgins et al., 2016) controls how close the distribution of bottleneck features $p(z|x = x_i)$ is to the prior $p(z)$, e.g., a standard Gaussian distribution independent of the sample identity $y = i$. So $\beta$ controls the strength of identity disentanglement on $z$. Lemma 1 shows that VAE is trained to minimize an upper bound of $-I(z, x) + \beta I(z, y)$, i.e., preserving most information of input $x$ in the bottleneck feature $z$ but removing from $z$ the critical information about sample identity $y = i$. Hence, the most identity-relevant information is disentangled from the VAE output $G(x) \sim p_D(x|z) = p(x|D(z))$ and preserved in the residual $R(x) \triangleq x - G(x)$.

### 3.3 IDENTITY-DISENTANGLED DATA GENERATION AND AUGMENTATION

We can study a data generative model based on identity-disentanglement of VAE, which generates $D(z)$ and $R(x)$ from $z$ and $y$ respectively and combine them to generate $x = G(x) + R(x)$, as shown in Fig. 3. The following lemma compares $I(R(x); y)$ with $I(x; y)$ and analyzes how much identity information can be preserved in $R(x)$.

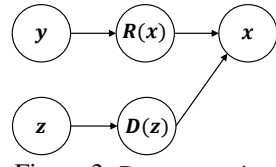

Figure 3: Data generation model.

**Lemma 2.** (Identity-disentangled data generation). *For a data generative model described above,*

$$I(R(x); y) \geq I(x; y) - I(z; y). \tag{7}$$

The detailed proof of Lemma 2 is given in the appendix.

**Theorem 1.** (Identity-disentangled data augmentation). *If we use a VAE in the identity-disentangled data generative model for Lemma 2, and if we define an augmentation $x' = R(x) + G'(x)$ with $G'(x) \sim q_D(x|z')$ and $z' = z + \delta$ (a $\delta$-perturbed $z$), we can lower bound $I(x'; y)$ as*

$$I(x'; y) \geq I(x; y) - \frac{1}{\beta}(L_{\text{VAE}} + I(z; x)). \tag{8}$$

Theorem 1 can be proven by combing Lemma 1 and Lemma 2, which is illustrated in the appendix.

Therefore, the augmentation $x'$ preserves most of the identity information in $x$ if $L_{\text{VAE}} + I(z; x)$ is small. VAE training aims at minimizing $L_{\text{VAE}}$ so the first term can be kept small. There is a trade-off between the second term and identity preservation $I(x'; y)$ since a sufficiently large $I(z; x)$ is necessary to produce augmentation $x'$ approximately drawn from the true data distribution. A key observation of the above theorem is that we can perturb $z$ to generate $x'$ without hurting the lower-bound of $I(x'; y)$. This implies that we can adversarially perturb $z$ to produce hard negatives and positives for more efficient contrastive learning without heavily distorting the original identity information. In contrast, most data augmentation techniques used in self-supervised learning have not taken this into account so they may change the sample-identity and result in poor representations.

### 3.4 IDENTITY-DISENTANGLED ADVERSARIAL AUGMENTATION (IDAA)

As discussed in Sec. 2 and Sec. 3.1, keeping a sufficient amount of hard positives and negatives in each batch $\mathbf{x}$ is critical to the effectiveness and sample efficiency of contrastive learning. Although any perturbation on $z$ can produce an identity-preserved data augmentation $x'$ according to Theorem 1, adversarially perturbing the VAE-bottleneck feature $z$ of the original samples can produce hard positives/negatives with sample-identities preserved. Meanwhile, we need to make sure that $x'$ is still close to the original data manifold, e.g., an natural image, so $x'$ needs to reside in the vicinity of $x$. Fortunately, we can use off-the-shelf adversarial attack algorithms for this purpose. The main difference here is: (1) we apply them to perturb the bottleneck features $z$ instead of $x$; and (2) the objective is to maximize the InfoNCE loss instead of a classification loss such as cross entropy. We call this method "identity-disentangled adversarial augmentation (IDAA)", as illustrated in Fig. 2.

In particular, the goal of IDAA is to generate an augmentation of $x$ in the form of $x'$ in Theorem 1 such that the InfoNCE loss is maximized using an $\epsilon$-bounded perturbation $\delta$ to $z = E(x)$ when generating $x'$. For a batch of data $\mathbf{x}$, this problem can be formulated as optimizing $\{\delta_i\}_{i=1}^N$ by

$$\max_{\|\delta_i\|_p \leq \epsilon, \, \forall i \in [N]} L_{\text{NCE}}(\mathbf{x}'), \quad x_i' = R(x_i) + D(E(x_i) + \delta_i). \tag{9}$$

To only perturb the positives and negatives to be harder for contrastive learning while keeping the anchors intact, we slightly modify $L_{\text{NCE}}(\mathbf{x}')$ to be

$$L_{\text{NCE}}(\mathbf{x}') = -\frac{1}{N} \sum_{i=1}^N \log \frac{\exp \langle f(x_i), h(x_i') \rangle}{\sum_{j=1}^N \exp \langle f(x_i), h(x_j') \rangle}, \tag{10}$$

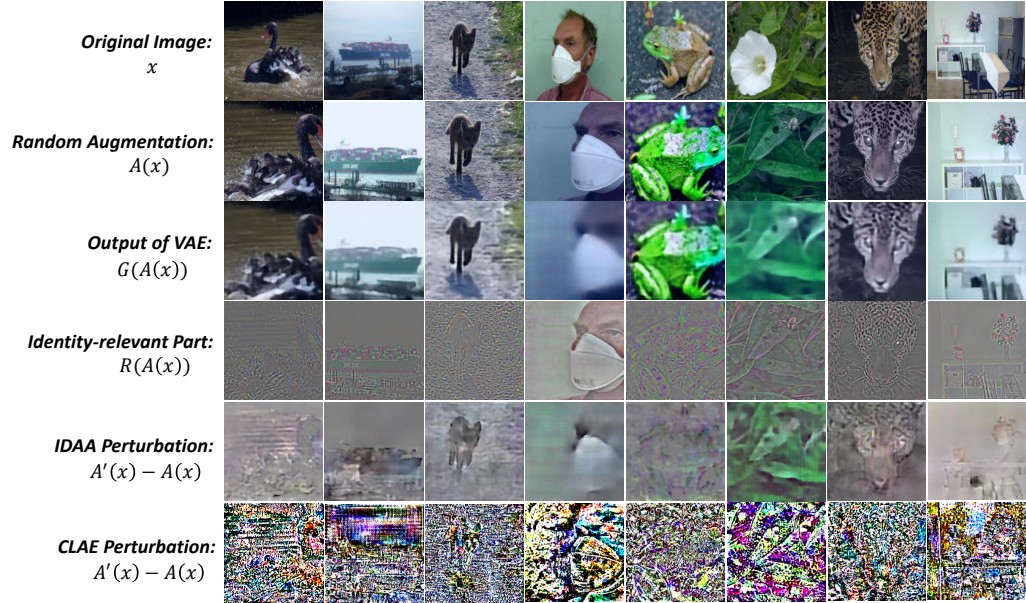

Original Image:
$x$

Random Augmentation:
$A(x)$

Output of VAE:
$G(A(x))$

Identity-relevant Part:
$R(A(x))$

IDAA Perturbation:
$A'(x) - A(x)$

CLAE Perturbation:
$A'(x) - A(x)$

Figure 4: Comparing IDAA and CLAE on their adversarial augmentations for ImageNet samples.

Since the augmentation $x'$ can be a positive for its own identification (i.e., when $x$ is the anchor) and a negative for other samples' identification, the objective in Eq. (9) modifies a sample $x$ to be both a hard positive and a hard negative. In other words, IDAA perturbs the bottleneck features to move $x'$ away from $x$ to other difference samples in the embedding space. Besides InfoNCE loss, IDAA can be applied to other self-supervised losses, e.g., replacing $L_{\text{NCE}}(\cdot)$ in Eq. (9) with the consistency regularization $L_{\text{CS}}(\cdot)$ in Eq (2).

IDAA is complementary to and can be applied to existing data augmentation techniques by replacing $x_i$ in Eq. (9) with a pre-defined data augmentation $A(x_i)$. It can also be applied together with other data augmentations, e.g., by combining multiple InfoNCE loss terms computed on different augmentations. A primary advantage of IDAA is that the generated augmentations are adaptive to the training models $f(\cdot)$ and $h(\cdot)$. It aims at finding the weaknesses of the self-supervised models on the sample-identification task and improve them without heavily distorting the original sample-identities, which is underexplored in previous literature.

There exists various off-the-shelf adversarial attack methods that can be directly applied to solve the problem in Eq. (9). In this paper, we adopt fast gradient sign method (FGSM) (Goodfellow et al., 2014) for its computational efficiency. FGSM perturbs $z = E(x)$ for one step by adding noises along the gradient sign's direction of the loss w.r.t. $\delta$ and the augmentation $x'$ is generated based on the perturbed bottleneck features $z + \delta$, i.e.,

$$x' = R(x) + D\left(E(x) + \delta^*\right), \quad \delta^* = \epsilon \, \text{sign}(\nabla_\delta L_{\text{NCE}}(\mathbf{x}')). \quad (11)$$

## 4 EXPERIMENTS

In this section, we evaluate the training efficiency and the test performance of several state-of-the-art methods in (1) self-supervised learning and (2) semi-supervised learning when using (1) their default random augmentations; (2) default augmentations + CLAE (Ho & Vasconcelos, 2020); and (3) default augmentations + IDAA. IDAA consistently improves both the training efficiency and the test accuracy of all the methods and outperforms the two baselines by large margins across several standard benchmark datasets. We also observe that IDAA trained on one dataset can be transferred to other datasets and improve downstream tasks on them, implying promising generalization capability of the learnable augmentation model. Moreover, we conduct a thorough sensitivity study of IDAA when changing (1) batch sizes; (2) network architectures; (3) training epochs; (4) regularization weight $\beta$ in the VAE objective; (5) dimensions of the VAE bottleneck features; and (6) adversarial attack strength bounded by $\epsilon$.

### 4.1 CASE STUDY OF AUGMENTATIONS GENERATED BY IDAA AND CLAE

We firstly present a case study of augmentations generated by IDAA, CLAE, and widely used random augmentations. We compare their identity preserving, hardness to contrastive learning, and the patterns of perturbations created by them on natural images. For random augmentations, we choose *RandomFlip, ColorJitter and GreyScale* for their popularity in recent SSL methods (Ye et al., 2019; Chen et al., 2020; Chen & He, 2021). In Fig. 1 (b)-(c), we compare the distance of an anchor to a positive with its distance to the nearest negative in the embedding space we apply the contrastive learning. If the latter is smaller than the former, i.e., the point is located below the "identity-preserving boundary" in the plots, its original identity is changed and cannot be preserved in the augmentation. On the other hand, if the latter is much larger than the former, i.e., the point is located on the left region of the plots, the sample identification task is too trivial and the contrastive learning is not efficient. From the plots, we can see that random augmentations are too easy while the CLAE augmentations are much harder but cannot preserve the original identity for some samples. In contrast, IDAA (ours) produces sufficiently hard augmentations within the boundary of identity changing, which is ideal for contrastive learning.

We visualize the augmentations produced by the three methods in Fig. 4 for natural images from ImageNet dataset. Both IDAA and CLAE are applied to random augmentations $A(x)$ and introduce further perturbations $A'(x) - A(x)$. IDAA generates more semantic perturbations to important regions of the images because its adversarial attack is conducted in the VAE bottleneck space, which is supposed to capture semantic attributes of the images. On the contrary, CLAE generates the perturbations on the input pixels and produces unnatural artifacts, which may distort the original sample-identity in the embedding space. In addition, we show the identity-relevant part $R(A(x))$ and the identity-disentangled part $G(A(x))$ produced by VAE. The former well preserves the most important patterns to identity the original sample, e.g., the edge of the steamship (column 2), spots of the frog (column 5) and pattern of the leopard (column 7). Since IDAA keep it intact in its augmentation model, the identity is well preserved in the augmentations.

### 4.2 SELF-SUPERVISED LEARNING

We evaluate data augmentations on four self-supervised learning methods: Plain (InfoNCE in Eq. (1)), UEL (Ye et al., 2019), SimSiam (Chen & He, 2021) and SimCLR (Chen et al., 2020) on three datasets, i.e., CIFAR10 (Krizhevsky et al., 2009), CIFAR100 (Krizhevsky et al., 2009), and miniImagenet (Vinyals et al., 2016). The experimental details are given in Sec. A.3 in the Appendix.

**Performance on Downstream Classification Tasks** We firstly evaluate the representations of different SSL methods using different data augmentations on downstream classification tasks. In Table 1, we report the test accuracy of classifiers produced by two commonly used evaluation protocols, i.e., k-nearest neighbor (kNN) with $k = 200$ and linear evaluation that trains a linear classifier on top of the learned representations. IDAA consistently improves the original SSL methods with their default random augmentations and the improvement is usually much larger than that brought by CLAE. In contrast, though CLAE achieves better or comparable accuracy than the default augmentations on most contrastive learning methods, it results in sig-

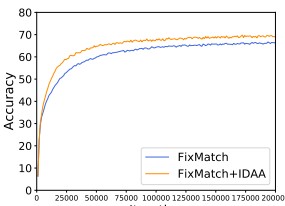

Figure 5: Convergence curve under 2500 labels on CIFAR100 for semi-supervised learning.

nificant degradation when applied to SimSiam, indicating that consistency regularization without comparison to negatives is more prone to potential identity distortion caused by adversarial augmentations in CLAE, which may strengthen the collapse of the training objective. Instead, IDAA exhibits better generalization to different types of SSL methods and evaluation protocols.

**Transfer Learning Performance** Different from random augmentations defined by human priors, IDAA and CLAE depend on models trained on a specific dataset. Hence, it is crucial to evaluate whether they can be transferred to other unseen datasets to improve downstream tasks. Following the linear evaluation protocol of Chen et al. (2020), we train encoders by IDAA and CLAE on ImageNet100 (i.e., 1oo classes subset of Imagenet) and then train linear classifiers on 8 datasets (Krizhevsky et al., 2009; Berg et al., 2014; Maji et al., 2013; Cimpoi et al., 2014; Parkhi et al., 2012; Nilsback & Zisserman, 2006; Welinder et al., 2010) using the encoders' output representations. The test accuracy of these classifiers are reported in Table 2. On Imagenet100, "SimCLR+IDAA"

| Method | kNN | | | Linear Evaluation | | |
|---|---|---|---|---|---|---|
| | CIFAR10 | CIFAR100 | miniImageNet | CIFAR10 | CIFAR100 | miniImageNet |
| Plain | 82.78±0.20 | 54.73±0.20 | 46.96±0.32 | 79.65±0.43 | 51.82±0.46 | 44.90±0.29 |
| Plain+CLAE | 83.09±0.19 | 55.28±0.12 | 47.01±0.28 | 79.94±0.28 | 52.14±0.21 | 45.43±0.15 |
| Plain+IDAA | **86.00±0.16** | **58.64±0.15** | **47.83±0.29** | **82.83±0.10** | **56.12±0.16** | **46.81±0.16** |
| UEL | 83.63±0.14 | 55.23±0.28 | 40.71±0.73 | 80.63±0.18 | 52.99±0.25 | 43.08±0.35 |
| UEL+CLAE | 84.00±0.15 | 55.96±0.06 | 41.75±0.39 | 80.94±0.13 | 54.27±0.40 | 44.32±0.24 |
| UEL+IDAA | **86.69±0.13** | **59.04±0.18** | **43.24±0.32** | **83.65±0.17** | **57.25±0.19** | **45.74±0.30** |
| SimSiam | 88.22±0.10 | 57.13±0.20 | 31.68±0.28 | 89.84±0.15 | 62.76±0.13 | 40.62±0.48 |
| SimSiam+CLAE | 85.59±0.21 | 53.88±0.08 | 27.77±3.47 | 87.77±0.08 | 60.89±0.22 | 37.32±0.47 |
| SimSiam+IDAA | **89.08±0.12** | **58.19±0.19** | **32.14±0.58** | **90.99±0.18** | **65.21±0.37** | **41.24±0.51** |
| SimCLR | 80.79±0.10 | 41.11±0.28 | 30.13±0.28 | 86.40±0.18 | 57.81±0.10 | 46.13±0.23 |
| SimCLR+CLAE | 80.27±0.18 | 43.57±0.17 | 32.23±0.08 | 85.25±0.07 | 57.69±0.25 | 46.76±0.16 |
| SimCLR+IDAA | **83.41±0.22** | **46.78±0.22** | **33.66±0.16** | **88.07±0.22** | **60.90±0.08** | **48.23±0.23** |

Table 1: Test accuracy on downstream classification: two evaluations of representations learned by four self-supervised learning methods using different data augmentations, i.e., their default ones, CLAE and IDAA (ours).

| | CIFAR10 | CIFAR100 | Birdsnap | Aircraft | DTD | Pets | Flower | CUB-200 |
|---|---|---|---|---|---|---|---|---|
| SimCLR | 61.83 | 36.55 | 12.68 | 24.19 | 54.35 | 46.46 | 75.00 | 16.73 |
| SimCLR+CLAE | 61.59 | 37.13 | 13.61 | 25.87 | 52.12 | 43.55 | 76.82 | 17.58 |
| SimCLR+IDAA | **64.49** | **38.82** | **13.89** | **26.02** | **54.97** | **46.76** | **77.99** | **18.15** |

Table 2: Transfer learning performance (test accuracy) on other datasets (trained on ImageNet100).

achieves 61.9% kNN accuracy, which outperforms 56.38% of SimCLR by a large margin. This improvement can be transferred to the 8 datasets. As shown in the Table, IDAA consistently outperforms CLAE and improves the original SimCLR, while CLAE brings degeneration on one dataset, i.e., Pets. We also compare their transfer learning results when applied to SimSiam in Sec. A.4 of the Appendix.

## 4.3 SEMI-SUPERVISED LEARNING

Data augmentation is also critical to state-of-the-art semi-supervised learning algorithms such as FixMatch (Sohn et al., 2020), which relies on accurate pseudo labeling and confidence-based data selection and their quality heavily depends on data augmentations. We apply IDAA to FixMatch to train a a WideResNet-28-8 model on CIFAR100 with only {400, 2500, 10000} labeled samples and the rest unlabeled. We train

| Method | CIFAR100 | | |
|---|---|---|---|
| | 400 labels | 2500 labels | 10000 labels |
| Fixmatch | 47.76 | 66.30 | 74.13 |
| Fixmatch+CLAE | 50.34 | 68.58 | 74.54 |
| Fixmatch+IDAA | **52.88** | **68.96** | **75.28** |

Table 3: Semi-supervised learning performance on CIFAR100 with different amounts of data labeled.

it for $2 \times 10^5$ steps with batch size of 512. The test accuracy of the trained models are reported in Table 3, where IDAA consistently improves FixMatch's accuracy and the improvement is more significant with fewer labeled data available. Moreover, as shown in Fig. 5, IDAA significantly improves the efficiency and convergence of FixMatch and can save a great amount of computation to reach a reasonable accuracy. IDAA keeps outperforming CLAE given different amounts of labeled data.

## 4.4 SENSITIVITY ANALYSIS OF HYPERPARAMETERS

**Batch size** We evaluate how SimCLR and SimCLR+IDAA perform using four different batch sizes and the results are shown in Fig. 6(a). As verified by previous works (Chen et al., 2020; He et al., 2020), increasing the batch size can improve SimCLR and other contrastive learning methods' performance, which is also reflected in our results. However, IDAA can significantly improve SimCLR's performance under small batch size, e.g., 64, because it effectively modifies various samples to be hard positives/negatives without distorting their original identities. Due to the same reason, SimCLR+IDAA is less sensitive to the change of batch size. It is also worth noting that SimCLR+IDAA using a very small batch size of 64 can surpass SimCLR with its best batch size of 512. This demonstrates the advantage of IDAA on improving the data/memory efficiency: with IDAA, contrastive learning is no longer limited to the usage of a large batch size.

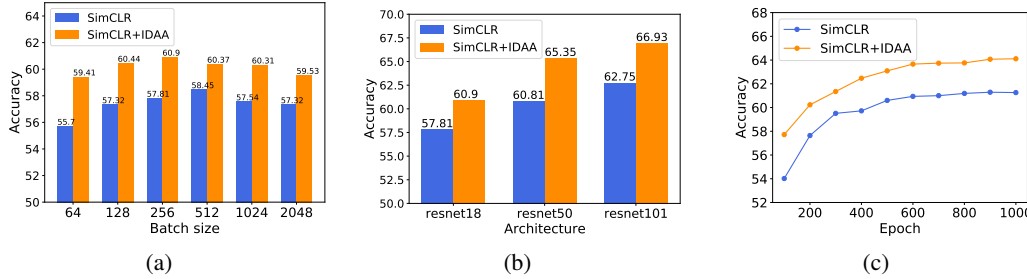

Figure 6: SSL performance under different (a) batch sizes, (b) ResNet architectures, and (c) training epochs.

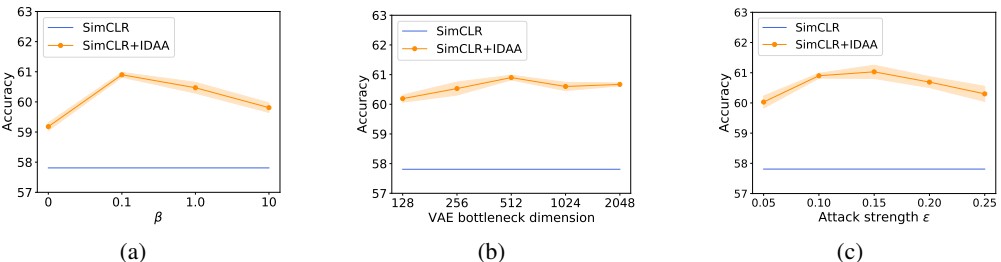

Figure 7: SSL performance using different (a) $\beta$, (b) VAE bottleneck dimensions, and (c) Attack strength $\epsilon$.

**Model Architecture**   We evaluate the linear evaluation performance of SSL when training different ResNet architectures in Fig. 6(b). Increasing the model size improves the performance of both methods but SimCLR+IDAA always outperforms SimCLR by a large margin. Hence, IDAA can improve SSL of different models and it can train a smaller model with less costs to match the performance of training a larger model by random augmentations.

**Training epochs**   As shown in Fig. 6(c), the performance of both methods improves when investing more training epochs but the SimCLR+IDAA saturates much earlier and only spends 300 epochs to achieve comparable performance as SimCLR trained using 1000 epochs. Hence, IDAA can greatly improve the training efficiency of SSL.

**Regularization weight $\beta$ in VAE**   As revealed by Theorem 1, $\beta$ in the VAE objective controls the lower bound of the identity information preserved in IDAA augmentations: larger $\beta$ enforces more identity preservation in $x'$ and stronger identity-disentanglement on $z$. However, it also controls $I(z; x)$ which reflects the proximity of $z$'s distribution to the true data manifold of $x$: large $\beta$ leads to small $I(z; x)$ and less information of $x$ preserved in $z$ that can be leveraged to produce stronger adversarial attacks for hard positives/negatives. In Fig. 7(a), we observe that the trade-off reaches a sweet spot at $\beta = 0.1$ among all the four $\beta$ values between 0 and 10.

**VAE bottleneck dimension**   As shown in Fig. 7(b), SSL performance with IDAA is not sensitive to the change of bottleneck dimension of VAE, though 512-dimension performs slightly better than other choices in the experiment.

**Attack strength $\epsilon$**   Stronger attacks may produce more hard positives/negatives but also increases the risk of identity distortion and unrealistic augmentations biased from the true data distribution/manifold. Results in Fig. 7(c) shows this trade-off and its effects on the SSL performance. Nevertheless, the performance of SimCLR+IDAA is still quite stable and only varies in a small range when changing $\epsilon$ since the identity-relevant information in $R(x)$ stays intact.

## 5   CONCLUSION

We propose a simple automatic data-augmentation method IDAA, which can generate more informative but identity-preserved augmentations to improve the efficiency and generalization of self-supervised learning. Motivated by an information theoretical analysis of VAE and identity preserving, IDAA adds adversarial noise to an identity-disentangled space learned by VAE and combines the perturbed VAE outputs with an intact identity-relevant part to produce augmentations. IDAA merely relies on a pretrained VAE without requiring any labeled data but consistently improves a diverse set of popular self-supervised/semi-supervised learning methods across multiple benchmarks. It also enhances the transfer learning performance and significantly improves the learning efficiency.

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

# A APPENDIX

## A.1 PROOF FOR PROPOSITION 1

**Proposition 1.** (Sample-identification likelihood as a lower bound of $I(x; y)$). *If $\mathbf{x}$ is a random mini-batch of size $N$ and the sample-identification likelihood of $x_i$ on its correct identification label $y = i$ to be $p(y = i | x = x_i)$, the mutual information $I(x; y)$ can be lower bounded by*

$$I(x; y) \geq \log N + \mathbb{E}_{\mathbf{x}} \left[ \frac{1}{N} \sum_{i=1}^{N} \log p(y = i | x = x_i) \right]. \tag{12}$$

*Proof.* To relate mutual information and sample-identification task, we take the expectation by first sample a mini-batch $\mathbf{x} = \{x_1, x_2, \cdots, x_N\}$ of size $N$ and then sample pairs $(x, y)$ from the mini-batch. Then we have:

$$
\begin{aligned}
I(x, y) &= H(y) - H(y|x) \\
&= \log N + \mathbb{E}_{\mathbf{x}} \left[ \sum_{y=1}^{N} \int \mathrm{d}x p(y, x) \log p(y|x) \right] \\
&= \log N + \mathbb{E}_{\mathbf{x}} \left[ \sum_{y=1}^{N} \int \mathrm{d}x p(x) p(y|x) \log p(y|x) \right] \\
&\geq \log N + \mathbb{E}_{\mathbf{x}} \left[ \sum_{y=1}^{N} \int \mathrm{d}x p(x) p(y|x) \log q(y|x) \right] \\
&= \log N + \mathbb{E}_{\mathbf{x}} \left[ \sum_{y=1}^{N} \int \mathrm{d}x p(y, x) \log q(y|x) \right] \\
&= \log N + \mathbb{E}_{\mathbf{x}} \left[ \sum_{y=1}^{N} \int \mathrm{d}x p(x|y) p(y) \log q(y|x) \right] \\
&= \log N + \mathbb{E}_{\mathbf{x}} \left[ \sum_{i=1}^{N} \int \mathrm{d}x p(x|y = i) p(y = i) \log q(y = i|x) \right] \\
&= \log N + \mathbb{E}_{\mathbf{x}} \left[ \sum_{i=1}^{N} \int \mathrm{d}x \delta(x - x_i) \frac{1}{N} \log q(y = i|x) \right] \\
&= \log N + \mathbb{E}_{\mathbf{x}} \left[ \sum_{i=1}^{N} \frac{1}{N} \log q(y = i|x = x_i) \right]
\end{aligned}
\tag{13}
$$

where $p(\cdot)$ denotes the true probability and $q(\cdot)$ denotes arbitrary estimation. The inequality in Equation (13) comes from $\mathrm{D}_{\mathrm{KL}}(p(y|x), q(y|x)) \geq 0$. □

## A.2 PROOF FOR THEOREM 1

We start by proving Lemma 2 and then we can prove Theorem 1 by combing Lemma 1 and Lemma 2.
**Lemma 2.** (Identity-disentangled data generation). *For a data generative model described above,*

$$I(R(x); y) \geq I(x; y) - I(z; y). \tag{14}$$

*Proof.* Due to the Markov chain $y \to (R(x), z) \to x$ in the data generative model in Fig. 3, we have

$$I(x; y) \leq I(R(x), z; y) = I(R(x); y) + I(z; y|R(x)). \tag{15}$$

By the definition of conditional mutual information, we have

$$I(z; y|R(x)) = I(z; y) + [H(R(x)|z) - H(R(x))] + [H(R(x)|y) - H(R(x)|y, z)]. \tag{16}$$

Since $z \perp\!\!\!\perp R(x)$ in the generative model, the last two terms in the above equation are zeros and $I(z; y|R(x)) = I(z; y)$. Substituting it to Eq. (15) completes the proof. □

|  | CIFAR10 | CIFAR100 | Birdsnap | Aircraft | DTD | Pets | Flower | CUB-200 |
|---|---|---|---|---|---|---|---|---|
| SimSiam | 49.75 | 25.59 | 7.32 | 16.29 | 40.07 | 31.33 | 58.20 | 9.96 |
| SimSiam+CLAE | 53.05 | 26.78 | 8.80 | 20.01 | 43.97 | 33.62 | 63.54 | 12.39 |
| SimSiam+IDAA | **55.74** | **29.63** | **9.16** | **20.16** | **47.32** | **36.22** | **65.63** | **12.84** |

Table 4: Transfer learning performance (in test accuracy %) on 8 datasets for CLAE and IDAA applied to SimSiam (pretrained on ImageNet100).

**Theorem 1.** (Identity-disentangled data augmentation). *If we use a VAE in the identity-disentangled data generative model for Lemma 2, and if we define an augmentation $x' = R(x) + G'(x)$ with $G'(x) \sim q_D(x|z')$ and $z' = z + \delta$ (a $\delta$-perturbed z), we can lower bound $I(x'; y)$ as*

$$I(x'; y) \geq I(x; y) - \frac{1}{\beta}(L_{\text{VAE}} + I(z; x)). \tag{17}$$

*Proof.* Applying the results from Lemma 2 and Lemma 1, we have

$$I(R(x) + G'(x); y) \geq I(R(x); y) \geq I(x; y) - I(z; y) \geq I(x; y) - \frac{1}{\beta}(L_{\text{VAE}} + I(z; x)). \tag{18}$$

The first inequality is due to the identifiability assumption extend from Theorem 1 in Robust PCA Candès et al. (2011), i.e., the identity-disentangled part $G(x)$ and the identity-revelant part $R(x)$ are separable give $x$. Under this assumption, we have the conditional entropy $H(R(x)|R(x)+G'(x)) = 0$ because $R(x) + G'(x)$ can be separated into $R(x)$ and $G'(x)$ again in an unique and exact way using VAE. This is a natural extension because VAE is an extension of robust PCA, as pointed out by Dai et al. (2018). □

### A.3 EXPERIMENTAL DETAILS OF SELF-SUPERVISED LEARNING

We evaluate data augmentations on four self-supervised learning methods: Plain (InfoNCE in Eq. (1), UEL (Ye et al., 2019), SimSiam (Chen & He, 2021) and SimCLR (Chen et al., 2020) on three datasets, i.e., CIFAR10 (Krizhevsky et al., 2009), CIFAR100 (Krizhevsky et al., 2009), and miniImagenet (Vinyals et al., 2016). The four SSL methods cover different contrastive learning methods (Plain, UEL, SimCLR) and a consistency regularization based method (SimSiam). The training/test splitting of miniImagenet follows Ebrahimi et al. (2020). Unless otherwise noted, a ResNet-18 with two separate BNs (Xie et al., 2020) for clean and adversarial samples is utilized as the representation encoder. The regularization weight $\beta$ in VAE is set to 0.1 and the bottleneck dimension is 512(3072) for CIFAR10/CIFAR100(miniImagenet).

### A.4 TRANSFER LEARNING PERFORMANCE USING SIMSIAM AS BASELINE

We further add new experiments of SimSiam, SimSiam+CLAE, and SimSiam+IDAA, and report their results in Table 4 as an extension of Table 2. Unlike SimCLR, SimSiam Chen & He (2021) adopts another popular idea of self-supervised learning based on the consistency regularization and Siamese network.

We evaluate the three methods on ImageNet-100 as we did for SimCLR, i.e., by running each method for 100 epochs and evaluating their transfer learning performance on the 8 datasets as in Table 2. Similar to SimCLR, IDAA consistently improves the transfer learning performance of SimSiam and outperform CLAE on all the datasets.

We select SimCLR and SimSiam for this study because they represent the two most widely studied self-supervised learning strategies, i.e., contrastive learning and consistency learning respectively.

## A.5 New Baseline: Data Augmentation without Decomposition ($G(z')$ only)

Here we compare with an additional baseline: simply attacking the z of a standard VAE (with beta=1) to produce an $x' = G(z')$ as augmentations for contrastive learning. We report its results on CIFAR in Table 5, denoted as "SimCLR+IDAA(w/o decomposition)". It shows that the performance significantly declines once the decomposition removed and it performs even worse than the original SimCLR.

Therefore, identity-disentanglement is critical and preserving the identity-relevant part $R(x)$ in augmentations is essential to self-supervised learning. This is illustrated in Figure 1. On the contrary, if we simply use the identity-distorted part $G(z')$ for augmentation with $R(x)$ removed, the positive and negative assignments in contrastive learning can be wrong. With many wrong identity labels, the identification task of contrastive learning can easily fail, resulting in poor representations and performance degradation on downstream tasks.

| Method | kNN | | Linear Evaluation | |
|---|---|---|---|---|
| | CIFAR10 | CIFAR100 | CIFAR10 | CIFAR100 |
| SimCLR | 80.79 | 41.11 | 86.40 | 57.81 |
| SimCLR+IDAA(w/o decomposition) | 67.25 | 32.77 | 75.82 | 47.41 |
| SimCLR+IDAA | **83.41** | **46.78** | **88.07** | **60.90** |

Table 5: Test accuracy on downstream classification tasks: comparing representations learned with $(x' = G(z') + R(x))$ and without identity-disentanglement decomposition $(x' = G(z'))$.

## A.6 Experiments on Lower Resolution version ($64 \times 64$) of ImageNet-1k

We add new experiments that train a ResNet-18 on ImageNet 64x64 by SimCLR, SimCLR+CLAE and SimCLR+IDAA. We run each method for 100 epochs with a batch size of 1024. We report the results in Table 6. It shows that SimCLR+IDAA still outperforms SimCLR+CLAE and original SimCLR on this more challenging dataset. Moreover, IDAA can improve both the kNN and linear evaluation accuracy while CLAE can only improve the kNN accuracy.

| | kNN | Linear Evaluation |
|---|---|---|
| SimCLR | 14.72 | 30.47 |
| SimCLR+CLAE | 15.44 | 28.77 |
| SimCLR+IDAA | **16.24** | **30.96** |

Table 6: Test accuracy of downstream classification on ImageNet $64 \times 64$.

## A.7 Comparison with more Baselines

In this section, we compare IDAA with two additional baselines published recently, i.e., InfoMin (Tian et al., 2020b) and Debiased (Chuang et al., 2020). InfoMin (Tian et al., 2020b) studies how to obtain good augmentation for contrastive learning, here we compare IDAA with InfoMin (Tian et al., 2020b) using their official code[1] on ImageNet they used in their paper (we reduced the resolution to 64×64 due to our limited computational resources). We apply each method to train ResNet-18 for 100 epochs and compare their linear evaluation accuracy on the test set. The results are reported in the Table 7, which shows that IDAA outperforms InfoMin (Tian et al., 2020b) by a large margin.

Debiased (Chuang et al., 2020) proposed a new objective, which aims to avoid sampling negatives of the same class as the anchor in contrastive learning. Our method as an augmentation strategy is complementary to Debiased . We add new experiments comparing with Debiased using their official

---

[1]https://github.com/HobbitLong/PyContrast

|  | Linear Evaluation |
|---|---|
| InfoMin (Tian et al., 2020b) | 27.11 |
| SimCLR+IDAA | **30.96** |

Table 7: Comparison with InfoMin for downstream classification on Imagenet $64 \times 64$.

code[2] on CIFAR10, the dataset used in their original paper. Specifically, we apply Debiased and "Debiased+IDAA" to train ResNet-18 for 300 epochs and compare their kNN an linear evaluation accuracy. The results are reported in Table 8, which shows that Debiased outperforms SimCLR but IDAA can further improve Debiased when applied on Debiased. Moreover, only "SimCLR+IDAA" can outperform Debiased by a large margin, i.e., 1.6% linear evaluation accuracy, which clearly demonstrate the effectiveness of IDAA.

|  | kNN | Linear Evaluation |
|---|---|---|
| SimCLR | 80.79 | 86.40 |
| Debiased(Chuang et al., 2020) | 81.96 | 86.47 |
| SimCLR+IDAA | 83.41 | 88.07 |
| Debiased(Chuang et al., 2020)+IDAA | **84.44** | **88.55** |

Table 8: Copmarison with Debiased for downstream classification on CIFAR10.

---

[2]https://github.com/chingyaoc/DCL

