# OpenReview forum: "Identity-Disentangled Adversarial Augmentation for Self-supervised Learning"
_ICLR.cc/2022/Conference — ICLR 2022 Submitted_

### Official Review · Reviewer_dMSm · 2021-10-29

**Correctness:** 3
**Technical Novelty And Significance:** 3
**Empirical Novelty And Significance:** 2
**Recommendation:** 5
**Confidence:** 4

**Main Review:**

## Strengths
+ The idea to decouple the identity from view generation seems to be novel and not explored in the CL literature before.
+ The presented method is simple and straightforward. I can see the potential of extending such formulation to a broader range of tasks and data domains.
+ The evaluation is relatively thorough, spanning across self-supervised & semi-supervised settings, and demonstrates the improvement over the vanilla and the CLAE method.

## Main Weaknesses
1. __Lack of comparisons to methods within the same category__
- The comparison is rather limited. The authors only compare to the vanilla (no adversarial augmentation introduced) and one method based on adversarial augmentation (CLAE). However, the basic idea is still _how to generate effective views / hard negative or positive samples_. Such problems have been extensively studied in the literature for CL [1-5]. How does IDAA perform against these methods? If not thoroughly evaluated, the performance gains are not justified and hence not interpretable.

2. __The definition of "identity-preserving"__
- In Fig.1, how is the distance measured? Is it measured in the feature space? How is the "Identity preserving boundary" defined? For CLAE, if the noise added to the input is bounded by a l_inf purturbation, then by definition the input should preserve the identity of the image. The information that this figure delivers is unclear.
- Continue on this point, to me it is very confusing on how you define identity-preserving. In fact, from Figure 4 in the paper, all augmentation methods actually preserve the identity. This also holds for CLAE. The only difference is that the perturbation generated by IDAA is more interpretable, but this has nothing to do with "identity-preserving". The augmented images still preserve their identities.
- Further, the motivation is not well explained. Given that the visualization does not directly reflect the point of "identity-preserving", the claims that current methods/augmentations "could change sample-identity and result in poor representations" is not justified.

3. __The IDAA is not self-contained as universal augmentation, but still needs a base set of augmentations__
- This might be a smaller issue, but IDAA still requires base augmentations (i.e., color jittering or random cropping/resizing) to work. In the experiments, all results of IDAA are based on some exsiting augmentations (e.g., those from SimCLR). The description of IDAA tends to sell it as a "universal" augmentation methods regradless of inputs, but domain knowledge (e.g., how you augment image data) is still needed from experiements. What is the actual performance of IDAA without using existing augmentations (those in SimCLR)?

## Other issues / questions
- Since the overall objective (i.e., generating better views for contrastive learning) is agnostic to the data modality, is the proposed scheme also be extended to other data domains beyond images, like time series / texts? What would be the most challenging parts when extending to other modalities?
- What is the actual computational cost (training time) compared to the vanilla SimCLR?

## References
[1] Hard Negative Mixing for Contrastive Learning.

[2] Adco: Adversarial contrast for efficient learning of unsupervised representations from self-trained negative adversaries.

[3] What makes for good views for contrastive learning.

[4] Debiased contrastive learning.

[5] Contrastive learning with hard negative samples.

**Summary Of The Paper:**

The paper studies contrastive self-supervised learning. In particular, it proposes an adversarial augmentation method for different view generation in contrastive learning. It claims that using such adversarial mechanism, the generated views are identity-preserving while being hard positives/negatives. Empirical experiments on several benckmarks validate the improved performance and efficiency.

**Summary Of The Review:**

Overall, the idea is interesting, however the current draft is not well-justified on the actual algorithm design, the motivation, and (perhaps the most important) empirical comparisons to existing CL methods that considers hard view mining. Further comments/questions are listed in the weaknesses / questions part.

The paper has its potential to the field, but issues need to be addressed first. I'm happy to change my score if the feedback addresses my concerns. Please refer to the points in the weaknesses / questions part. I would like to see feedbacks on these comments/questions.

---

> ### Author Response · Authors · 2021-11-22
> **Response to Reviewer dMSm: Part I**
>
> We appreciate your time and suggestions! We have added new experimental comparisons with two additional baselines in the revised version of our paper. Here are our detailed replies to your questions.
>
> **(1) Lack of comparisons to methods within the same category: The comparison is rather limited. The basic idea is still how to generate effective views / hard negative or positive samples. Such problems have been extensively studied in the literature for CL [1-5].**
>
> - We address a different problem as [1] and [5]: they focus on **data selection**, e.g., how to **select hard negatives**, while we focus on **data augmentation**, i.e., how to **generate more informative yet identity-preserved augmentations for self-supervised learning**. Hence, our contribution is **complementary** to that of [1] and [5], e.g., one can use IDAA for data augmentation followed by [1] or [5] selecting harder examples from our augmentations.
>
> - Both AdCo [2] and InfoMin [3] study data augmentation. AdCo[2] adopts a very similar idea as CLAE, i.e., applying adversarial attack algorithms to generate augmentation. Since we already compared IDAA with CLAE thoroughly in our main paper, in the new experiments, we focus on comparing with InfoMin [3] using their official code on ImageNet they used in their paper (we reduced the resolution to 64$\times$64 due to our limited computational resources). We apply  each method to train ResNet-18 for 100 epochs and compare their linear evaluation accuracy on the test set. The results are reported in the table below, which shows that **IDAA outperforms InfoMin [3]** by a large margin.
>
> |Imagenet 64x64|Linear Evaluation|
> |:--|:--|
> |InfoMin[3]|27.11|
> |SimCLR+IDAA|**30.96**|
>
> - Our method as an augmentation strategy is complementary to the objective proposed in Debiased [4], which aims to avoid sampling negatives of the same class as the anchor in contrastive learning. We add new experiments comparing with Debiased [4] using their official code on CIFAR10, the dataset used in their original paper. Specifically, we apply Debiased [4] and "Debiased+IDAA" to train ResNet-18 for 300 epochs and compare their kNN an linear evaluation accuracy. The test accuracy for them is reported in the table below: Debiased [4] outperforms SimCLR but applying IDAA in Debiased [4] (Debiased+IDAA) can bring **further improvement**. Moreover, even without using Debiased, "SimCLR+IDAA" can still **outperform Debiased [4] by 1.6\%** on the linear evaluation accuracy, which demonstrates the consistent advantage brought by IDAA when combined with different methods.
>
> |CIFAR10|kNN|Linear Evaluation|
> |:--|:--|:--|
> |SimCLR|80.79|86.40|
> |Debiased[4]|81.96|86.47|
> |SimCLR+IDAA|83.41|88.07|
> |Debiased+IDAA|**84.44**|**88.55**|

---

> > ### Author Response · Authors · 2021-11-22
> > **Response to Reviewer dMSm: Part II**
> >
> > **(2) The definition of "identity-preserving"**
> >
> > **(2.1) In Fig.1, how is the distance measured? Is it measured in the feature space? How is the "Identity preserving boundary" defined? For CLAE, if the noise added to the input is bounded by a $l _ \infty$ purturbation, then by definition the input should preserve the identity of the image. The information that this figure delivers is unclear.**
> >
> > - As illustrated by line 5-7 in the first paragraph of Sec 4.1 and Fig.1's caption, the distance is measured in the **representation space** we apply the contrastive learning. The "Identity preserving boundary" is where a sample is **equally distant to its positive(s) and to its nearest negative**, so the neural network cannot distinguish the positive from the negative. If crossing the boundary and reaching a point below the Identity preserving boundary in Fig. 1, the sample becomes **closer to the negative than to the positive**. This indicates that the data augmentation changes the original identity and contrastive learning will learn from overly distorted data.
> >
> > - Although CLAE restricts the perturbation by an $l _ p$-norm bound $\epsilon$, it **may still change some samples' identity** (unless $\epsilon=0$) because **samples differ in their distances to the boundary** and an uniform $l _ p$-norm bound cannot guarantee identity-preserving for every sample, e.g., the ones near the boundary.
> >
> > **(2.2) Continue on this point, to me it is very confusing on how you define identity-preserving. In fact, from Figure 4 in the paper, all augmentation methods actually preserve the identity.**
> >
> > - With all due respect, we believe that you misunderstood the meaning of "identity-preserving" and thus misinterpreted Figure 4. Theoretically, an augmentation $x'$ of a sample $x$ preserves $x$'s identity $y$ **if $I(x';y)\geq I(x;y)-\epsilon$ holds with a small $\epsilon$, e.g., Eq. (8) for IDAA**. Empirically, "identity-preserving" refers to that a sample is **closer to positive(s) than to its nearest negative** in contrastive learning. As shown in Fig. 1 (b)-(c) (please also see the x-axis and y-axis names), many CLAE augmentations cannot preserve the original identity since they produce negatives closer to the anchor than the positives.
> >
> > - Since augmentations including CLAE and IDAA are limited to make too much change to the original samples ($\epsilon$-ball constraint for adversarial perturbations in CLAE and IDAA), **it is hard for human to notice whether the sample identity is changed or not for the neural nets**. This is similar to adversarial attacks: they change a neural network's predictions of images using perturbations that are too small to be noticed by human eyes, because **neural nets usually rely on very sparse patterns to make predictions**, as discussed in [6].
> >
> > **(2.3) Further, the motivation is not well explained. Given that the visualization does not directly reflect the point of "identity-preserving", the claims that current methods/augmentations "could change sample-identity and result in poor representations" is not justified.**
> >
> > As demonstrated in Fig. 1 (b)-(c) and discussed in Sec 4.1, with more explanations above, CLAE is likely to change the identity of some augmentations since **some are outside (below) the identity preserving boundary**. On the contrary, IDAA can preserve identity information well: most points of IDAA are **within (above) the identity preserving boundary**. Moreover, a variety of empirical results in Sec 4.2 and 4.3 thoroughly demonstrate that IDAA results in better representations than existing augmentations such as CLAE.

---

> > > ### Author Response · Authors · 2021-11-22
> > > **Response to Reviewer dMSm: Part III**
> > >
> > > **(3) The IDAA is not self-contained as universal augmentation, but still needs a base set of augmentations: it still requires base augmentations (i.e., color jittering or random cropping/resizing) to work.  What is the actual performance of IDAA without using existing augmentations (those in SimCLR)?**
> > >
> > > - We respectfully disagree with your opinion. IDAA is proposed as a complementary technique rather than an alternative to existing random augmentations. Random augmentations have been broadly used in almost all tasks. In fact, self-supervised learning methods such as **SimCLR carefully fine-tune or search for the best random augmentation operators**. Hence, there is no reason not using them！The question is: can we **further improve self-supervised learning on top of these "optimal" random augmentations**? This is exactly what we addressed in this paper. As demonstrated by thorough experiments, we show that IDAA is complementary to these random augmentations since IDAA can still bring significant improvement on top of them.
> > >
> > > - **Without any random augmentation applied in SimCLR, IDAA can still significantly improves the performance of SimCLR**, though the performance is much poorer than that with random augmentations. This is demonstrated by the results on CIFAR100 reported in the table below.
> > >
> > > ||kNN|Linear Evaluation|
> > > |:--|:--|:--|
> > > |SimCLR (w/o random augmentation)|21.16|34.71|
> > > |SimCLR+IDAA (w/o random augmentation)|**27.67**|**42.44**|
> > >
> > > **(4) Since the overall objective (i.e., generating better views for contrastive learning) is agnostic to the data modality, is the proposed scheme also be extended to other data domains beyond images, like time series / texts? What would be the most challenging parts when extending to other modalities?**
> > >
> > > **The proposed scheme and its theoretical insights are principal and can be extended to other data domains**: the VAE based identity-disentanglement and the identity-preserved adversarial augmentation can be directly applied to other domains such as time series/texts.
> > > One of the most challenging parts when extending to other modalities is choosing the **proper space to conduct identity-disentanglement**. For example, on NLP data, we may perform identity-disentanglement in an embedding space instead of the raw discrete input space and choose transformer as the architecture for VAE's encoder and decoder.
> > >
> > >
> > > **(5) What is the actual computational cost (training time) compared to the vanilla SimCLR?**
> > >
> > > - Although IDAA requires more computation per epoch caused by the extra inference and adversarial perturbation on VAE, it produces **more informative augmentations** (more challenging but identity-preserved) that can **significantly reduce the number of training epochs needed to reach the same accuracy**. In practice, it can effectively reduce the overall training time.
> > >
> > > - For example, in our experiments on CIFAR100, the (averaged) training time per epoch on CIFAR100 (batch size=256) is 67.6s (seconds) for "SimCLR+IDAA" and 35.4s for vanilla SimCLR. However, as reported in the table below, IDAA **greatly saves the total training time to reach a similar performance**. For example, to reach >60% accuracy, SimCLR takes $1.77\times 10^4$s while IDAA taks only $1.35\times 10^4$s and thus saves over $4\times 10^3$s training time.
> > >
> > > ||SimCLR||SimCLR+IDAA||
> > > |:--|:--|:--|:--|:--|
> > > |Epoch|Training time (s)|Linear Evaluation|Training time (s)|Linear Evaluation|
> > > |100|$3.54\times 10^3$|54.02|$6.76\times 10^3$|57.72|
> > > |200|$7.08\times 10^3$|57.64|$1.35\times 10^4$|60.23|
> > > |300|$3.54\times 10^4$|59.51|$2.02\times 10^4$|61.35|
> > > |400|$1.42\times 10^4$|59.72|$2.70\times 10^4$|62.47|
> > > |500|$1.77\times 10^4$|60.60|$3.38\times 10^4$|63.09|
> > > |600|$2.12\times 10^4$|60.94|$4.06\times 10^4$|63.66|
> > > |700|$2.48\times 10^4$|61.00|$4.73\times 10^4$|63.74|
> > > |800|$2.83\times 10^4$|61.19|$5.41\times 10^4$|63.76|
> > > |900|$3.19\times 10^4$|61.29|$6.10\times 10^4$|64.07|
> > > |1000|$3.54\times 10^4$|61.26|$6.76\times 10^4$|64.11|
> > >
> > > ***
> > >
> > > [1] Hard Negative Mixing for Contrastive Learning.
> > >
> > > [2] Adco: Adversarial contrast for efficient learning of unsupervised representations from self-trained negative adversaries.
> > >
> > > [3] What makes for good views for contrastive learning.
> > >
> > > [4] Debiased contrastive learning.
> > >
> > > [5] Contrastive learning with hard negative samples.
> > >
> > > [6] Andrew Ilyas, Shibani Santurkar, Dimitris Tsipras, Logan Engstrom, Brandon Tran, Aleksander Madry. Adversarial Examples Are Not Bugs, They Are Features, NeurIPS 2019.

---

> > > > ### Comment · Reviewer_dMSm · 2021-11-26
> > > > **Concerns remained**
> > > >
> > > > I thank the authors for their detailed feedback and the newly added experiments. However, I still have several concerns regrading the response.
> > > >
> > > > __[Comparison to other methods.]__
> > > >
> > > > First, to me the principle behind these methods are the same: how to generate effective views. Even if you claim your method is complementary to [1] and [5], the most direct method is to conduct an ablation study that they are really orthogonal; otherwise it is not justifiable.
> > > >
> > > > Further, the setting for comparison to InfoMin seems wired -- you only train 100 epochs, which can not even guarantee convergence. Usually people also use larger model as backbone rather than ResNet-18. The new results are not satisfying and to me, do not justify IDAA is better than these methods.
> > > >
> > > > __[On identity-preserving.]__
> > > >
> > > > I'm still not convinced by the response on "identity-preserving". First of all, identity-preserving should be defined w.r.t. human interpretation, rather than neural nets. The author uses adversarial examples as an example for identity-preserving. However, this example directly demonstrates why "identity-preserving" should be w.r.t. human evaluation: with $l_p$ bounded noise, neural nets produce highly confident but wrong predictions, whereas the image identity does not change (for human).
> > > >
> > > > I feel the argument for "identity-preserving" here is not tightly associated with the actual performance. In the input space, by definition, all these methods / augmentations should genearate identity-preserving samples in order to preserve information for downstream tasks. How the feature space (distance in the feature space) evolves during training that relates to the final downstream accuracy, however, is not showed in the paper.
> > > >
> > > > __[Base augmentation in SimCLR & extending to other modalities.]__
> > > >
> > > > Unfortunately, I'm not convinced by the experiments comparing SimCLR vs. SimCLR+IDAA without base augmentations.
> > > >
> > > > First, contrastive learning requires effective augmentations to work. The augmentations in SimCLR are not "random", but on the contrary, carefully designed for visual data. In SimCLR, if no "Color Jitterring" applied for CL, it will easily learn a shortcut solution which identifies the color distribution (see experiments in the SimCLR paper).
> > > >
> > > > Based on this, I'm also less convinced that the framework can be extended to other modalities where we don't have augmentation priors. As the "SimCLR+IDAA (w/o SimCLR aug)" experiment shows, IDAA alone suffers from large performance drop.
> > > >
> > > > The fact that "Developing an augmentation method on top of existing augmentations" just limits the significancy and applicability of IDAA, let alone only visual data being used.
> > > >
> > > > __[Computation costs.]__
> > > >
> > > > The new experiments validated my concern -- IDAA is much computationally expensive than the vanilla SimCLR. It does not make sense to compare under the same time budget when they are not even converged. Once you are comparing the final performance, they should be based on the converged performance, where you can claim IDAA brings accuracy gains, but not more efficient.

---

> > > > > ### Author Response · Authors · 2021-11-26
> > > > > **[Response 1/2] Thank you for your further response! But most of them are groundless, incorrect, or contradictory in many ways!**
> > > > >
> > > > > We appreciate the reviewer for the further comments! However, with all due respect, **most of them are groundless or inaccurate**: (1) the reviewer may not get the problem we target in this paper, which is **"augmentation for self-supervised learning" in our paper title**; (2) the reviewer's **understanding (guess) of the key concept "identity-preserving" in this paper is fundamentally wrong! It is defined for neural nets representations
> > > > > rather than human visual cortex!** (3) the reviewer's **opinions and critics are contradictory with each other!** Here are our detailed reply to those concerns.
> > > > >
> > > > > ```
> > > > > (1) Comparison to other methods
> > > > > ```
> > > > >
> > > > > - **The current comparison to other methods is sufficient**: we applied IDAA to **THREE different contrastive learning methods** (Plain, UEL and SimCLR), **TWO different non-contrastive learning methods** (SimSiam and FixMatch) on **11 datasets** (CIFAR10, CIFAR100, miniImageNet, ImageNet 64 $\times$ 64, Aircraft, DTD, Pets, Flower and CUB-200), and **IDAA consistently brings more significant improvement than other widely-used or recently-proposed data augmentation baselines** (random augmentation, CLAE and InfoMin [3]).
> > > > >
> > > > > - **IDAA (our method) does NOT belong to the category of [1] and [5]**, which obviously do not study data augmentation generation (the problem of our paper and CLAE [7]) but study hard negative mining (**please read the paper titles of [1] and [5] and compare them with ours**). Instead, we study instance-adaptive data augmentation for SSL as CLAE[7] so we followed CLAE's experiments to **thoroughly compare with all the baselines that CLAE compared in their original paper [7]**. In addition, **we added more methods to apply IDAA to (SimSiam[8] and FixMatch[9]) or compare IDAA to (InfoMin[3]), which were not included in CLAE paper [7]**.
> > > > >
> > > > > - Due to the same reason above, we train a ResNet-18 for 100 epochs on ImageNet, which follows **exactly the same setting in CLAE [7]. So this comparison is Correct and Fair!** Moreover, **InfoMin [3] also studies data augmentation for SSL and their paper trains models for only 200 epochs** in most experiments.
> > > > >
> > > > > - We will try our best to run experiments on [1] and [5], and InfoMin [3] with larger model and more epochs. But training larger model on ImageNet requires much more computational resources and we are not sure we can complete them before the end date of the discussion phase.
> > > > >
> > > > >
> > > > > ```
> > > > > (2) On identity-preserving
> > > > > ```
> > > > >
> > > > > We believe the reviewer fundamentally misunderstood our ''identity-preserving'', which has been explained in line 15-16 in Sec.3.3, line 5-7 in Sec 4.1, and many places in the paper. Identity preservation is for neural nets representations instead of for the visual cortex of human. **Please read the sentence below Eq. (3): $p(y|x)$ here is not a model of human brain**, it is approximated by $q_{NCE}$ (Eq. (1)) defined by a neural net $f(\cdot)$. If you have time, please also read Theorem 1 and Eq. (8), which give a concrete example of "identity preserving" used in this paper.
> > > > >
> > > > > - "identity-preserving should be defined w.r.t. human interpretation" ---**This is incorrect!** In most self-supervised learning settings widely used nowadays, **there is no human intervention in the training process of a neural network**.
> > > > >
> > > > > - As pointed out by [6] and many other works, neural networks mainly rely on a very sparse region of an image in producing the prediction. **This is very different from human cognition**. In this paper, we study "identity-preserving" for neural nets not for human.
> > > > >
> > > > > - **The case study of "identity-preserving" based on neural nets representation in Fig. 1 is consistent with the downstream task performance**. Fig. 1(b)-(c) show that IDAA augmentations can preserve the identity information better than CLAE augmentations because it retains more points within the boundary. This explains why IDAA-equipped SSL outperforms CLAE-equipped SSL in all the downstream tasks.

---

> > > > > > ### Author Response · Authors · 2021-11-26
> > > > > > **[Response 2/2] Thank you for your further response! But most of them are groundless, incorrect, or contradictory in many ways!**
> > > > > >
> > > > > > ```
> > > > > > (3) Base augmentation in SimCLR & extending to other modalities.
> > > > > > ```
> > > > > >
> > > > > > - **If we already have prior knowledge of pre-defined augmentation operators or the optimal augmentation found in SimCLR, there is no reason for not using them!** Therefore, the problem of "Developing an augmentation method on top of existing augmentations" (e.g., SimCLR+IDAA) makes more sense and has more practical value than "framework can be extended to other modalities where we don't have augmentation priors."
> > > > > >
> > > > > > "Developing an augmentation method on top of existing augmentations" is **the standard setting adopted by related works and major baselines that also study data augmentation for SSL**, e.g., CLAE [7] and InfoMin [3], and we follow their setting. CLAE (InfoMin) do not remove SimCLR (CMC) augmentations from SimCLR (CMC) when applying their augmentations.
> > > > > >
> > > > > > "Developing an augmentation method on top of existing augmentations" is not trivial but even more challenging, given that the "existing augmentations" have already been optimized, carefully designed, or are the best one, e.g., SimCLR finds the best augmentation among 49 candidates. As demonstrated by thorough experiments in this paper, **IDAA can consistently and significantly improve SimCLR with the "optimal" augmentations**.
> > > > > >
> > > > > > - On the other hand, **if we do not have any prior knowledge of which augmentation operators to use for a modality**, which is highly unlikely in reality, **IDAA can still generate informative augmentations to improve the performance. This is shown in our new experiments requested by you**. We argue this is not the common case, though we did the new experiments for your previous question: "What is the actual performance of IDAA without using existing augmentations (those in SimCLR)?"
> > > > > >
> > > > > > - **If we know some candidate augmentation operators but do not know the optimal ones**, we can use a brute-force search similar to SimCLR or we can train a model to do this job like AutoAugment, and then we run IDAA on top of the best choice found by these off-the-shelf methods. However, this is out of the major scope of the problem we study here, i.e., the problem studied in CLAE [7] and InfoMin [3].
> > > > > >
> > > > > > - You said "I'm also less convinced that the framework can be extended to other modalities where we don't have augmentation priors." **Note we have not claimed this is our target. The new experiments are requested in your comments and we quote it here:** "What is the actual performance of IDAA without using existing augmentations (those in SimCLR)?" **Your critic here also contradicts with your statement** of "contrastive learning requires effective augmentations to work". If we start from contrastive learning with no effective augmentation, it can easily fail and we cannot learn any meaningful data augmentation model from its training process.
> > > > > >
> > > > > >
> > > > > > ```
> > > > > > (4) Computation costs.
> > > > > > ```
> > > > > >
> > > > > > - **We are not sure what you would like to compare here: computational costs or performance at convergence? SimCLR+IDAA performs better on both metrics.**
> > > > > >
> > > > > > - If it is the former, wall-clock time reported in our new results is the gold criterion and SimCLR+IDAA spends much less wall-clock time to reach the same accuracy as vanilla SimCLR, demonstrating that IDAA reduces the computational costs. For example, vanilla SimCLR reaches its best accuracy $61.29$ at epoch-900 and spends $3.19\times 10^4$ seconds, while SimCLR+IDAA reaches a better accuracy $61.35$ at epoch-300 and spends only $2.02\times 10^4$ seconds.
> > > > > >
> > > > > > - If it is the latter, the table shows that the accuracy at convergence for vanilla SimCLR is $61.29$, which is lower than $64.11$ of SimCLR+IDAA, demonstrating that IDAA improves the accuracy at convergence.
> > > > > >
> > > > > > - In summary, **SimCLR+IDAA is more efficient than SimCLR to reach SimCLR's best accuracy, and SimCLR+IDAA's accuracy at convergence is also much higher than that of vanilla SimCLR.** Therefore, IDAA improves both the efficiency and performance at convergence.
> > > > > >
> > > > > > ***
> > > > > > [1] Hard Negative Mixing for Contrastive Learning.
> > > > > >
> > > > > > [2] Adco: Adversarial contrast for efficient learning of unsupervised representations from self-trained negative adversaries.
> > > > > >
> > > > > > [3] What makes for good views for contrastive learning.
> > > > > >
> > > > > > [4] Debiased contrastive learning.
> > > > > >
> > > > > > [5] Contrastive learning with hard negative samples.
> > > > > >
> > > > > > [6] Andrew Ilyas, Shibani Santurkar, Dimitris Tsipras, Logan Engstrom, Brandon Tran, Aleksander Madry. Adversarial Examples Are Not Bugs, They Are Features, NeurIPS 2019.
> > > > > >
> > > > > > [7] Chih-Hui Ho, Nuno Vasconcelos. Contrastive Learning with Adversarial Examples, NeurIPS 2020.

---

> ### Author Response · Authors · 2021-11-30
> **NEW experiments requested by Reviewer dMSm: Comparisons to two more baselines and Experiments on larger models.**
>
> As you suggested, we (1) **compared our data augmentation method (IDAA) with a hard negative mining method as you requested (HCL [5])** ([1] has not released their code); and (2) added **new experiments of IDAA on larger models (ResNet-50) and comparison to InfoMin [3] as you requested**.
>
> The results show that (1) IDAA is **complementary to hard negative mining (i.e., further improving HCL); and (2) IDAA still outperforms InfoMin [3] (another data augmentation method) when training larger models**. Here is our detailed reply.
>
> ```
> (1) Compare our data augmentation method with hard negative mining methods [1][5]
> ```
>
> Here we compare our method with hard negative mining methods [1][5]. Since [1] has not released their code, we mainly focus on comparison with HCL [5]. We **follow HCL's original setting** to train a ResNet50 with 400 epochs on CIFAR using (1) HCL only; and (2) HCL+IDAA. The results are reported in the table below, which shows:
>
> - IDAA is **complementary to hard negative mining method** like HCL and **further improves HCL** on two metrics, i.e., kNN and linear evaluation, over two datasets, i.e., CIFAR10 and CIFAR100.
>
> - Although HCL is already a very powerful strategy for contrastive learning that achieves 91.48\% accuracy on CIFAR10, IDAA can still bring significant improvement to it and  **achieves 93.10\% accuracy on CIFAR10**. This demonstrates that **IDAA is a highly effective and general data augmentation strategy that can improves different SoTA methods.**
>
> ||CIFAR10||CIFAR100||
> |--:|:--|:--|:--|:--|
> ||kNN|Liner Evaluation|kNN|Linear Evaluation|
> |HCL|89.03|91.48|60.46|68.83|
> |HCL+IDAA|**90.88**|**93.10**|**64.21**|**71.81**|
>
> ```
> (2) Compare our data augmentation method (IDAA) with a recent data augmentation method (InfoMin [3]) on a lager model.
> ```
> Here we **follow the original setting of InfoMin [3]** to train a ResNet-50 by contrastive learning (1) using InfoMin [3]; and (2) using IDAA on ImageNet (images resized to 64$\times$ 64 due to the time and computation limit). It will take days on the maximum number of GPUs we can afford. The training has ran for 100 epochs and we report the current results in the table below. Our goal is to run it for 200 epochs, i.e., the same number of epochs used in InfoMin [3] paper, and we will report the new results once we get them. According to previous experiences on ImageNet (64$\times$ 64), the difference between 100 epochs and 200 epochs is not phenomenal.
>
> The results show that IDAA outperforms InfoMin [3] by a large margin on training ResNet-50, i.e., **improving its top-1 accuracy of linear evaluation by >1.5\%.**
>
> |ImageNet 64$\times$ 64|Linear Evaluation|
> |--:|:--|
> |InfoMin[3]|36.44|
> |IDAA|**38.28**|
>
> We believe that we addressed all your concerns and completed all the experiments you requested with positive results. Please let us know if you still have more concerns. Thanks!

---

> ### Author Response · Authors · 2021-11-30
> **We addressed all your concerns with new experiments. Would you mind confirming if you have further concerns and reconsidering your rating? Thanks!**
>
> Dear Reviewer dMSm:
>
> We appreciate your comments and time! We addressed all your concerns with detailed responses below. Would you mind checking them and confirm if you have further questions? Here is a short summary:
>
> - We compared our data augmentation method (IDAA) with a **hard negative mining method** as you requested (HCL [5]) ([1] has not released their code); and added new experiments of IDAA **on larger models (ResNet-50) and comparison to InfoMin [3]** as you requested.
>
> - We **clarified your misunderstanding on "identity-preservation"**: it is defined for neural nets representations instead of for human visual cortex.
>
> - **Studying augmentations on top of existing augmentations**: this is a common setting **adobted by ALL augmentation methods for SSL, i.e., CLAE, AdCo and InfoMin**. They did **NOT** remove the augmentation of SimCLR/CMC when applying their augmentations and **we followed this standard setting for a fair comparison**.
>
> - **Computation cost**: the wall-clock time comparison (the gold criterion for evaluation of  computation cost) between "SimCLR+IDAA" vs. SimCLR showed that IDAA cam **improve both the efficiency and effectiveness** at covergence.
>
> Would you mind letting us know if you have any further concerns? Thanks a lot!
>
> Best Regards,
>
> Authors

---

### Official Review · Reviewer_uy6S · 2021-11-01

**Correctness:** 4
**Technical Novelty And Significance:** 4
**Empirical Novelty And Significance:** 4
**Recommendation:** 6
**Confidence:** 4

**Main Review:**

Strengths:

- The idea is novel;
- The main hypothesis that other augmentation strategies result in pair of samples that are either too easy or run the risk of corrupting the instance-level information is backed up by empirical evidence;
- This work provides a information theory based interpretation of the IDAA strategy with sufficient proofs;
- The claim that better hard positives/negatives result in more sample-efficient models w.r.t the batch size and the number of epochs is highlighted through various ablations;
- IDAA provides significant performance improvements on all reported benchmarks;

Weaknesses:

- Skipping the conclusion is not an acceptable practice to gain more space. One could move some of the training hyperparameters to the appendix to gain some extra space in the main paper for that;
- The ablation on the batch size raises the concern that IDAA does not scale up too while. It is unclear whether SimCLR will surpass SimCLR+IDAA with a sufficiently large minibatch. Indeed, we observe that increasing the batch size is beneficial for SimCLR only on Figure 6-a;
- The datasets explored in the experiments of this work are somewhat limited in terms of diversity. Training a contrastive self-supervised model on the full imagenet dataset can be resource intensive, however the authors could have explored the use of lower resolution versions of it while keeping all the classes (e.g. ImageNet 64x64);

Minor improvements:
- Typo in the introduction: 'ullustrated in Fig. 2'

**Summary Of The Paper:**

This work introduces a novel adversarial augmentation strategy. This augmentation, nammed Identity-Disentangled Adversarial Augmentation (IDAA) aims at producing better hard positives/negatives while preserving the identity of the sample. This is achieved by perturbing the latent space of a variational auto-encoder while preserving the identity relevant features through the residual term. This allows for the training of better and more sample efficient self/semi-supervised.

**Summary Of The Review:**

This work introduces a novelty augmentation strategy with sufficient empirical and theoretical arguments. Although some concerns remain about its scalability to larger batch sizes and more challenging datasets, the authors clearly show significant performance improvements when using IDAA across all experimental settings and empirically justify all claims/hypotheses made.

---

> ### Author Response · Authors · 2021-11-22
> **Response to Reviewer uy6S:**
>
> We appreciate your time and suggestions! We have added new experiments of larger batch sizes and experiments on ImageNet 64$\times$64 in the new version of our paper. We have modified the paper according to your comments. Here are our detailed replies to your questions.
>
> **(1) Skipping the conclusion is not an acceptable practice to gain more space.**
>
> As you suggest, we will add a conclusion section.
>
> **(2) It is unclear whether SimCLR will surpass SimCLR+IDAA with a sufficiently large minibatch.**
>
> - The performance does not monotonically improve as the batch size increases. In fact, contrastive learning usually has a limitation of requiring large batch size because hard negatives are very rare in random mini-batches so only a large batch can cover sufficient hard negatives and avoid collapse. In IDAA, we address this problem by an adversarial perturbation method that changes the easy positives/negatives to hard ones without distorting their identities. Therefore, SimCLR+IDAA can achieve better performance than SimCLR even using a smaller batch size. This demonstrates that IDAA improves the data efficiency of contrastive learning because large batch size increases the memory cost and computation overhead.
>
> - Moreover, we add new experiments of larger batch sizes of 1024 and 2048 on CIFAR100 and report the results in the table below. It empirically verifies that keep increasing the batch size does not improve the performance of SimCLR: the sweet spot is batch size = 512 and the performance drops after that. In addition, it shows that SimCLR+IDAA consistently outperforms SimCLR by a large margin under larger batch sizes. It is also worth noting that SimCLR+IDAA using a very small batch size of 64 can surpass SimCLR with its best batch size of 512, which demonstrates the advantage of IDAA on improving the data/memory efficiency. With IDAA, contrastive learning is no longer limited to using a large batch size.
>
> |Batch Size|64|128|256|512|1024|2048|
> |:--|:--|:--|:--|:--|:--|:--|
> |SimCLR|55.70|57.32|57.81|58.45|57.54|57.32|
> |SimCLR+IDAA|**59.41**|**60.44**|**60.90**|**60.37**|**60.31**|**59.53**|
>
> **(3) The datasets explored in the experiments of this work are somewhat limited in terms of diversity. The authors could have explored the use of lower resolution versions of it while keeping all the classes (e.g. ImageNet 64x64)**
>
> As you suggest, we add new experiments that train a ResNet-18 on ImageNet 64x64 by SimCLR, SimCLR+CLAE and SimCLR+IDAA. We run each method for 100 epochs with a batch size of 1024. We report the results in the table below. It shows that SimCLR+IDAA still outperforms SimCLR+CLAE and original SimCLR on this more challenging dataset. Moreover, IDAA can improve both the kNN and linear evaluation accuracy while CLAE can only improve the kNN accuracy.
>
> ||kNN|Linear Evaluation|
> |:--|:--|:--|
> |SimCLR|14.72|30.47|
> |SimCLR+CLAE|15.44|28.77|
> |SimCLR+IDAA|**16.24**|**30.96**|

---

> > ### Comment · Reviewer_uy6S · 2021-11-25
> > **Response to Authors' feedback.**
> >
> > Thank you for your feedback. This addresses well my initial concerns.
> >
> > I do have an additional question. I was under the impression that Figure 1.c was computed using some unrelated features (e.g. VGG, INv3, etc) but you stated in your reply to reviewer dMSm that the features are obtained through contrastive learning. If by that you mean that these are the features obtained by using IDAA training then this undermines the conclusions you draw from it. Is this the case?

---

> > > ### Author Response · Authors · 2021-11-25
> > > **Response to Reviewer uy6S's further comments:**
> > >
> > > Thank you for your feedback! **We are glad to hear that our response has addressed all your initial concerns!** Here is our detailed reply to your new concerns.
> > >
> > > ```
> > > If by that you mean that these are the features obtained by using IDAA training then this undermines the conclusions you draw from it. Is this the case?
> > > ```
> > >
> > > - **The model used to compute the distance in Figure 1(c) is neither trained using IDAA nor CLAE**: it is **trained using the original SimCLR before convergence** because we aim at **simulating the intermediate stage of self-supervised learning** when the model does not fully converge and when high-quality data augmentations with identity preserved are critical to the future training (while poor augmentations with identity distorted are detrimental to the future training).

---

### Official Review · Reviewer_Augv · 2021-11-01

**Correctness:** 2
**Technical Novelty And Significance:** 3
**Empirical Novelty And Significance:** 3
**Recommendation:** 5
**Confidence:** 3

**Main Review:**

Finding better augmentation schemes for self-supervised learning is undoubtedly and important and challenging problem. While the proposed method does seem to offer empirical gains with respect to baselines, I am not completely convinced about the motivation and some of the theoretical claims.

In particular, in Theorem 1, the authors claim that they can perturb z to produce x' without hurting the lower bound. However, it seems odd that this theorem does not have a dependency on $\delta$ (that is used to perturb z). After all, one could construct a z' that is arbitrarily different from z (using a large delta). In this case, x' could be very different from x and I(x', y) would be very small (possibly violating the lower bound). The proof utilizes I(R(x) + G'(x); y) ≥ I(R(x); y), without explaining why this is the case. Overall, I do not see how this is true for arbitrary $\delta$.

Other comments:

* The claims made in Section 4.1 regarding the qualitative differences between augmentation samples (in Figure 4) are subjective and hand-wavy. Based on visual inspection alone, the difference in quality between these augmentations is not apparent and thus should be removed.
* Baselines:
    - Why do the authors only consider SimCLR in Table 2?
    - In Table 3, why are CLAE and random augmentations missing?
    - I am curious whether simply attacking the z of a standard VAE (with beta=1) to produce an x' = G(z') would lead to vastly different performance. In general, I am not convinced that the decomposition into the generated and residual components is necessary. This seems like an essential baselines to compare to.
* All the plots in the paper need to be made larger and more legible.

**Summary Of The Paper:**

This paper proposes a novel augmentation method for contrastive SSL. To this end, the authors leverage VAEs—generating an augmentation for an image x by adversarially perturbing it's encoding (z = E(x)) to generate a perturbed reconstruction (G(z')), which is then added back to the original residual (R(x) = x - G(x)). The motivation for this approach is to maintain the identity preserving information via R(x), while creating challenging augmentations. The authors empirically evaluate their approach for downstream classification, transfer learning and semi-supervised learning.

**Summary Of The Review:**

While the empirical results of this paper are interesting, I still have concerns about the theoretical claims and the comparison to baselines. I would be happy to increase my score if the authors addressed these.


#### Post-rebuttal Update ####

I thank the authors for their detailed response. Based on the additional experimental results that the authors presented to all the reviewers, I have decided to increase my score to a 6. I still have two comments:

- The authors' response regarding theorem 1 does not completely address my concern. Theorem 1 as stated right now allows z to be perturbed by an arbitrary $\delta$, regardless of x. (In contrast, in adversarial training x itself is perturbed by $\delta$ which is constrained by some $\epsilon$.) Thus, it is still not clear how the bound in Theorem 1 can hold for some arbitrary $\delta$. For instance, imagine we have two images $x_1$ and $x_2$ and their corresponding latent $z_1$ and $z_2$. I could simply define $z_1' = z_2$ by selecting $\delta = z_2 - z_1$. In the case, I would expect that the bound is violated. I believe that there should either be an assumption on $\delta$, or it should feature in the resulting bound.

- In the caption for Table 8, there is a typo: `copmaring`.

- The authors should incorporate all the additional results they reported during the rebuttal into the paper (or appendix).


#### Post-discussion Update ####

Over the course of the discussion between the authors and the reviewers, it has become apparent that several of the experimental settings chosen in the paper are non-standard, which make it hard to verify the validity of the results. I do share the concerns of my fellow reviewers in this regard, and thus will downgrade my score to a 5.

---

> ### Author Response · Authors · 2021-11-22
> **Response to Reviewer Augv: Part I**
>
> We appreciate your time and comments! We have added more explanation to Theorem 1 and compared our method with the baselines you suggested in the new version of the paper. We have modified the paper according to your comments.
>
> **(1)  However, it seems odd that this theorem does not have a dependency on $\delta$  (that is used to perturb $z$). After all, one could construct a $z'$ that is arbitrarily different from $z$ (using a large delta). In this case, $x'$ could be very different from $x$ and $I(x', y)$ would be very small (possibly violating the lower bound). The proof utilizes $I(R(x) + G'(x); y) \geq I(R(x); y)$, without explaining why this is the case. Overall, I do not see how this is true for arbitrary $\delta$.**
>
> - Applying a large $\delta$ in IDAA cannot arbitrarily change $x'$ from $x$ due to several reasons: (1) batch normalization layers in VAE decoder limits the scale of $G'(x)$; (2) adversarial attack algorithms always enforce a bounded $\delta$; (3) VAE decoder is trained to produce $G(x)$ close to $x$ due to the reconstruction error in its training objective. Therefore, $G'(x)$ still lies in a reasonable range and will not dominate $R(x)$ in $x'\triangleq R(x)+G'(x)$.
>
> - The above reasons validate an identifiability assumption, i.e., the identity-disentangled part $G(x)$ and the identity-relevant part $R(x)$ are separable given $x$. Under this assumption, we have the conditional entropy $H(R(x)|R(x)+G'(x))=0$ because $R(x)+G'(x)$ can be separated into $R(x)$ and $G'(x)$ again in an unique and exact way using VAE. This assumption is extended from Theorem 1 in Robust PCA [1], which claims that a matrix can be separated into a low-rank part and a sparse part in a unique and exact way. This is a natural extension because VAE is an extension of robust PCA, as pointed out by [2].
>
> **(2) Based on visual inspection alone, the difference in quality between these augmentations is not apparent and thus should be removed.**
>
> The claims in Section 4.1 are mainly based on the comparison between the perturbations generated by CLAE and IDAA shown in Figure 4, where the latter shows clearly more semantic patterns than the former. That being said, we agree that showing the augmentations produced by CLAE and IDAA in Figure 4 is not necessary due to the small magnitude of their perturbations (a result of the $\epsilon$-ball constraint of adversarial attacks). Hence, we will remove them from Figure 4.
>
> **(3) All the plots in the paper need to be made larger and more legible.**
>
> Thanks for the advice! we will make all the plots larger and more legible.

---

> > ### Author Response · Authors · 2021-11-22
> > **Response to Reviewer Augv: Part II**
> >
> > **(4) Baselines**
> >
> > **(4.1) Why do the authors only consider SimCLR in Table 2?**
> >
> > We further add new experiments of SimSiam, SimSiam+CLAE, and SimSiam+IDAA, and report their results in the table below as an extension of Table 2. Unlike SimCLR, SimSiam [3] adopts another popular idea of self-supervised learning based on the consistency regularization and Siamese network.
> >
> > We evaluate the three methods on ImageNet-100 as we did for SimCLR, i.e., by running each method for 100 epochs and evaluating their transfer learning performance on the 8 datasets as in Table 2. Similar to SimCLR, IDAA consistenly improves the transfer learning performance of SimSiam and outperforms CLAE on all the datasets.
> >
> > We select SimCLR and SimSiam for this study because they represent the two most widely studied self-supervised learning strategies, i.e., contrastive learning and consistency learning respectively.
> >
> > ||CIFAR10|CIFAR100|Birdsnap|Aircraft|DTD|Pets|Flowers|CUB-200|
> > |:--|:--|:--|:--|:--|:--|:--|:--|:--|
> > |SimSiam|49.75|25.59|7.32|16.29|40.07|31.33|58.20|9.96|
> > |SimSiam+CLAE|53.05|26.78|8.80|20.01|43.97|33.62|63.54|12.39|
> > |SimSiam+IDAA|**55.74**|**29.63**|**9.16**|**20.16**|**47.32**|**36.22**|**65.63**|**12.84**|
> >
> > **(4.2) In Table 3, why are CLAE and random augmentations missing?**
> >
> > The original FixMatch already includes a carefully designed and selected set of random augmentations that have been tuned to achieve the best performance.
> > We newly add the experiment of "FixMatch+CLAE" on CIFAR100 to Table 3 and report the results in the table below. IDAA consistently outperforms CLAE in semi-supervised learning with different numbers of labeled data. It is worth noting that IDAA shows more advantage in less labeled data case, e.g., IDAA surpasses CLAE by over 2\% where only 400 labeled data is avaliable.
> >
> > ||400 labels|2500 labels|10000 labels|
> > |:--|:--|:--|:--|
> > |FixMatch|47.76|66.30|74.13|
> > |FixMatch+CLAE|50.34|68.58|74.54|
> > |FixMatch+IDAA|**52.88**|**68.96**|**75.28**|
> >
> > **(4.3) I am curious whether simply attacking the z of a standard VAE (with beta=1) to produce an x' = G(z') would lead to vastly different performance.**
> >
> > We have tested the performance of simply attacking the $z$ of a standard VAE (with beta=1) to produce an $x' = G(z')$ as augmentations for contrastive learning. We report its results on CIFAR in the table below, denoted as "SimCLR+IDAA(w/o decomposition)". It shows that the performance significantly declines once the decomposition removed and it performs even worse than the original SimCLR.
> >
> > Therefore, identity-disentanglement is critical and preserving the identity-relevant part $R(x)$ in augmentations is essential to self-supervised learning. This is illuestrated in Figure 1. On the contrary, if we simply use the identity-distorted part $G(z')$ for augmentation with $R(x)$ removed, the positive and negative assignments in contrastive learning can be wrong. With many wrong identity labels, the identification task of contrastive learning can easily fail, resulting in poor representations and performance degradation on downstream tasks.
> >
> > ||kNN||Linear Evaluation||
> > |:--|:--|:--|:--|:--|
> > ||CIFAR10|CIFAR100|CIFAR10|CIFAR100|
> > |SimCLR|80.79|41.11|86.40|57.81|
> > |SimCLR+IDAA(w/o decomposition)|67.25|32.77|75.82|47.41|
> > |SimCLR+IDAA|**83.41**|**46.78**|**88.07**|**60.90**|
> >
> > ***
> >
> > [1] Emmanuel J. Candes, Xiaodong Li, Yi Ma, and John Wright. Robust Principal Component Analysis, https://arxiv.org/abs/1711.01558.
> >
> > [2] Bin Dai, Yu Wang, John Aston, Gang Hua, David Wipf. Connections with robust PCA and the role of emergent sparsity in variational autoencoder models, JMLR 2018.
> >
> > [3] Xinlei Chen and Kaiming He. Exploring simple siamese representation learning, CVPR 2021.

---

> ### Author Response · Authors · 2021-11-25
> **Response to Reviewer Augv's further comments:**
>
> Thank you for your feedback and support! Here are our detailed replies to your further comments:
>
>
> **(1) It is still not clear how the bound in Theorem 1 can hold for some arbitrary $\delta$. For instance, imagine we have two images $x_1$  and $x_2$ and their corresponding latent $z_1$ and $z_2$. I could simply define $\delta$ by selecting $\delta=z_2-z_1$. In the case, I would expect that the bound is violated.**
>
>
> - In A.2 of the Appendix, we made an identifiability assumption extended from Theorem 1 of Robust PCA [1], i.e., the identity-disentangled part $G(x)$ and identity-relevant part $R(x)$ of $x$ are unique and separable using VAE. **Under this assumption, the bound is NOT violated in your example, and here is why**: In your example of $\delta$, if we set $z _ 2'=z _ 2+\delta=z _ 1$, then the augmentation for $x _ 2$ is $x _ 2'= G(x _ 1)+R(x _ 2)$. Under the identifiability assumption, we can use VAE to disentangle $x_2'$ into $G(x _ 1)$ and $R(x _ 2)$ again in an unique and exact way, i.e., $G(x _ 1)=G( G(x _ 1)+R(x _ 2) )$ and $R(x _ 2) = G(x _ 1)+R(x _ 2)-G( G(x _ 1)+R(x _ 2) )$. Hence, we have $H(R(x _ 2 )| G(x _ 1)+ R(x _ 2) )=0$, which leads to $I(G(x _ 1)+ R(x _ 2);y) \geq I(R(x _ 2 );y)$.
>
> - **The identifiability assumption is a mild and reasonable assumption** because VAE is known as an extension of Robust PCA [1], as pointed out by [2]. Similar assumption can be found in Proposition 1 of [3], which assumes the reversibility of Autoencoders.
>
> - In IDAA, **due to the $\epsilon$-ball constraint of the adversarial perturbation, your example of $\delta$ will almost never happen** because the norm of $\delta=z_2-z_1$ for two different images can easily exceed the commonly used small $\epsilon$. This also indicates that the identifiability assumption holds in practice with high probability.
>
> **(2) In the caption for Table 8, there is a typo: copmaring.**
>
>
> - Thanks for capturing that! We will fix it in the next version.
>
>
> **(3) The authors should incorporate all the additional results they reported during the rebuttal into the paper (or appendix).**
>
>
> - We will follow your suggestion to incorporate all the additional results reported during the rebuttal into the paper (or appendix).
>
> ***
>
> [1] Emmanuel J. Candes, Xiaodong Li, Yi Ma, and John Wright. Robust Principal Component Analysis, https://arxiv.org/abs/1711.01558.
>
> [2] Bin Dai, Yu Wang, John Aston, Gang Hua, David Wipf. Connections with robust PCA and the role of emergent sparsity in variational autoencoder models, JMLR 2018.
>
> [3] Xun Huang, Ming-Yu Liu, Serge Belongie, Jan Kautz. Multimodal Unsupervised Image-to-Image Translation, ECCV 2018.

---

> ### Author Response · Authors · 2021-11-29
> **Can you provide the reasons for downgrading the score? We have addressed ALL you concerns! The setting IS standard between baselines!**
>
> Would you mind elaborating on your reasons for downgrading the score after we addressed all your concerns? We have addressed all your concerns with plenty of new empirical evidence and theoretical analysis!
>
> - **Our experimental setting IS STANDARD in previous works (CLAE [7], InfoMin [3]) studying the same problem!** The problem is data augmentation for self-supervised learning and we simply follow their settings! Can you read these papers before judging " it has become apparent that several of the experimental settings chosen in the paper are non-standard"？**This is NOT FAIR!**
>
> - **To answer your question, we added THREE NEW groups of experiments! And in total, l we reported SEVEN NEW groups of experiments to answer all reviewers' concerns!** These experiments well addressed all the concerns and provide strong evidence that our method excels in different settings the reviewers requested. Would you mind sharing your reason for not being convinced by them?
>
> We appreciate your time and comments! However, we respectfully disagree with your decision of downgrading without discussing the new concerns leading to the downgrade. Would you mind adding more details about the reasons behind it? Thanks！

---

> > ### Comment · Area_Chair_N42t · 2021-11-29
> > **AC's comments**
> >
> > Dear Authors,
> >
> > Each reviewer makes his/her assessment based on not only the authors' rebuttal but also the other Reviewers' comments and discussions. It is uncommon for a reviewer to reassess his/her position as the discussion phase continues. Note the average score is a useful but not decisive criterion when the AC makes the final recommendation to the SAC and PCs.
> >
> > Best,
> >
> > AC

---

### Official Review · Reviewer_kfHb · 2021-11-02

**Correctness:** 4
**Technical Novelty And Significance:** 3
**Empirical Novelty And Significance:** 3
**Recommendation:** 6
**Confidence:** 2

**Main Review:**

This work studies an important area of machine learning that is important to a vast literature in self-supervised learning. In particular, they propose a method that utilizes the strengths of deep variational autoencoders and their information theoretic properties to inherit identity disentanglement.

The main strengths of the paper is the utility and simplicity of such a method. As illustrated on the extensive applications, the method is readily applicable to schemes and I see the work presented in this paper to be extremely useful for the machine learning community at large for applications in data augmentation beyond those described in this work. In particular, this work may be useful for adversarial training via augmentation, which is a closely related area and given the 'adversarial' motivation hinted in this paper, can be developed into such a scheme.

The simplicity of the paper also serves to some extent as a weakness. This is due to the fact that there are several questions that may be asked albeit not taking too much merit away from the idea. The first is how specific is this method to the VAE? While the main reason for using VAEs is the link to information theory (the appearance of mutual information), which is the reason the main theorems work, I wonder how applicable this result would be at least heuristically with other deep generative methods such as the Wasserstein Autoencoder for example. I think such avenues can be deferred to future work however I would be interested to know the author's position on this.

The paper is very well presented and I found the level of exposition very high with little or no issues understanding the contribution of this work. I am a non-expert in this area however one small question that does emerge is the motivation of using an adversary for this task. While I understand the adversary takes the role to strengthen the identity preservation, I do feel that this paper may be conflated with adversarial training, which as mentioned earlier is a potential application yet not explored. This is however a minor point as a non-expert, it may be a common phrase used for disentanglement.

**Summary Of The Paper:**

This work presents a new technique for disentanglement based inspired by the premise of an adversary that generate diverse samples that preserve identities via augmentation. The main motivation of the paper is for self-supervised learning which is an important area that benefits a lot from data augmentation. In particular, they use the reconstructed sample of a VAE and its distorted difference (G(x) - x) as an adversary to perform data augmentation and show its effectiveness via success on a number of different self-supervised learning tasks.

**Summary Of The Review:**

I believe the paper presents a new method and serves to a useful problem of machine learning for a number of tasks. Due to the simplicity and utility of the method, I lean towards acceptance of the paper.

---

> ### Author Response · Authors · 2021-11-22
> **Response to Reviewer kfHb:**
>
> We appreciate your time and suggestions! Here are our detailed replies to your questions.
>
> **(1) The first is how specific is this method to the VAE?  I wonder how applicable this result would be at least heuristically with other deep generative methods such as the Wasserstein Autoencoder for example.**
>
> From both theoretical and empirical intuitions, we believe that VAE is a simpler but better choice than other deep generative models. The main reason is that our objective here is not reconstruction/generation but identity-disentanglement and VAE serves best for this purpose:
>
> - **VAE provides tighter bound for identity-disentanglement** in Lemma 1. For example, Wassertein Autoencoder (WAE) [1] matches the ***marginal distribution*** of latent factor $p _ E(z)$ to the prior $p(z)$, while VAE matches the ***conditional distribution*** $p _ E(z|x)$ to prior $p(z)$. By the convexity of KL divergence, we have $I(z;y) \leq {\rm E} _ x\left[{\rm D} _ {\rm KL}( p _ E\left(z|x\right) \|\| p(z))\right] \leq {\rm D} _ {\rm KL}( {\rm E} _ x \left[p _ E(z|x) \right] \|\| p(z)) = {\rm D} _ {\rm KL}( p _ E(z) \|\| p(z))$. Hence, comparing to WAE, VAE optimizes a tighter bound for the identity-disentanglement measured by $I(z;y)$.
>
> - Empirically, other deep generative model like **WAE or GAN may generate higher-quality reconstruction** $G(x)$ than VAE so their residual **$R(x)=x-G(x)$ tends to preserve less identity information**, which may lead to more identity distortion in generating the augmentations and hence performance degradation.
>
> **(2) The motivation of using an adversary for this task. While I understand the adversary takes the role to strengthen the identity preservation, I do feel that this paper may be conflated with adversarial training.**
>
> - The adversary **does not mean to strengthen the identity preservation**, which is instead guaranteed by perturbation in the VAE bottleneck space, no matter whether the perturbation is adversartial or not. The motivation of using an adversary in IDAA is to **generate hard negatives/positives**, which is essential to the **efficiency** of contrastive learning [2-4], as most existing methods using random positives/negatives usually need very large batch size to cover sufficient number of hard ones.
>
> - Since our adversarial perturbation is applied to the VAE **bottleneck features** while most adversarial attacks are applied to the **input pixels**, we are not tackling the same problem as previous adversarial training methods, whose goal is to improve the robustness to the input-space attacks.
>
> ***
>
> [1] Ilya Tolstikhin, Olivier Bousquet, Sylvain Gelly, Bernhard Schoelkopf. Wasserstein Auto-Encoders, ICLR 2017.
>
> [2] Ting Chen, Simon Kornblith, Mohammad Norouzi, and Geoffrey Hinton. A simple framework forcontrastive learning of visual representations. ICML 2020.
>
> [3] Kaiming He,  Haoqi Fan,  Yuxin Wu,  Saining Xie,  and Ross Girshick.   Momentum contrast forunsupervised visual representation learning.  CVPR 2020.
>
> [4] Xinlei Chen and Kaiming He. Exploring simple siamese representation learning. CVPR 2021.

---

### Official Review · Reviewer_rByQ · 2021-11-24

**Correctness:** 3
**Technical Novelty And Significance:** 3
**Empirical Novelty And Significance:** 3
**Recommendation:** 5
**Confidence:** 4

**Main Review:**

The idea to disentangle the identity with VAE is technically correct and the analysis in section 3 with the information bounds and ELBO provides a clear theoretical support method. The proposed IDAA seems to be able to improve the current augmentation strategy by removing the identity attributes when generating augmentations. A weakness here is IDAA seems to rely upon existing pre-defined augmentation combinations. The authors better tune down their claim or provide more support that IDAA can achieve on par performance even with random augmentation or without any augmentation.

Although I like the analysis in section 3, I still have some concerns about the claim that IDAA can improve different SSL methods. As known recent SSL methods can be generally classified as contrastive and non-contrastive ones. The contrastive methods, such as SimCLR, mainly use InfoNCE loss, and the analysis in section 1 well supports the relation of IDAA to these methods. However, the relation to non-contrastive methods like SimSiam, SWAV, etc. is not clear since they do not use InfoNCE loss. Thus, I am curious whether IDAA really improves the performance of these methods. If the authors would like to claim IDAA generally improves existing SSL methods, I think it is also necessary to show how IDAA works with recent SSL methods such as MoCo, SWAV, BYOL, DINO, etc.

I also have two main questions about the experiment part:

- The results reported in the paper seem not consistent with existing papers in SSL area. For example, SimCLR and SimSiam are known to achieve over 90% accuracy in linear classification when ResNet18 backbone with 800 epochs, and the results in Table 1 seem to have significant differences. In Table 1 CLAE seems to harm the performance of SimCLR and SimSiam, which is not consistent with the conclusion in their paper either. Based on these results, it is hard to judge whether IDAA improves existing SSL methods' performance and outperforms other augmentation strategies.

- According to $\beta$-VAE, the disentanglement is affected by choice of $\beta$, and they have better disentanglement in the latent space when $\beta$ is bigger than 1. According to Eqn (5) in the paper, it also seems the identity is better preserved from wrong augmentation when $\beta$ is bigger (please correct me if I am wrong). The result in Fig. 7 seems to suggest the best choice is $\beta=0.1$, which is a little bit out of my expectation and I am worried this could violate the lower bound shown in Eqn (8). Could the authors provide additional justifications with some analytical examples or provide some possible explanation?

**Summary Of The Paper:**

This work presents an augmentation strategy leveraging the latent space of VAE and adversarial samples, named Identity-Disentangled Adversarial Augmentation (IDAA). IDAA perturbs the latent space of a pretrained variational auto-encoder to produce adversarial samples that preserve the identity attributes. Experiments show that IDAA improves the performance of existing SimCLR and SimSiam baselines.

**Summary Of The Review:**

This paper provides a novel augmentation strategy that leverages the latent space of pretrained VAEs to produce adversarial samples. The technique part is generally correct but is limited to InfoNCE loss, and the experiment results are not completely consistent with existing papers. Based on these concerns, I had difficulty in judging the value of the proposed method. I am currently inclined to recommend a reject and I suggest the authors keep the same configuration of the baselines to justify the improvement of their method.

---

> ### Author Response · Authors · 2021-11-25
> **[Response 1/2] Thank you for the comments! But most of them are Incorrect or Inaccurate.**
>
> We appreciate the reviewer for the detailed comments! **However, with all due respect, most of them are incorrect or inaccurate**. We believe that the reviewer ignore or misinterpret some of our theoretical and empirical results. Here are our detailed reply to those concerns.
>
> ```
> (1) However, the relation to non-contrastive methods like SimSiam, SWAV, etc. is not clear since they do not use InfoNCE loss. If the authors would like to claim IDAA generally improves existing SSL methods, I think it is also necessary to show how IDAA works with recent SSL methods such as MoCo, SWAV, BYOL, DINO, etc.
> ```
>
> - **Our theoretical analysis holds not only for the InfoNCE loss but also for all the contrastive learning methods** (which cover a rich class of self-supervised learning approaches) such as MoCo [1] mentioned in your comments, as long as they estimate $q(y|x)$ (the conditional distribution identity $y$ given sample $x$). This is illustrated in line 17 of Appendix A.1. In InfoNCE, the estimation is $q_{NCE}$ in Eq. (1). Other contrastive learning methods use different estimations, e.g., MoCo estimates $q(y|x)$ using an dictionary queue.
>
> - **Our theoretical motivation of identity preservation also holds for consistency/Siamese-type self-supervised learning methods** such as SiaSiam and BYOL mentioned in your comments: they aim at maximizing the consistency between the augmentations and thus keeping their identity to be the same. However, our theory cannot be rigorously applied to them due to their representation collapse in the worst case, i.e., they can trivially achieve a minimal consistency loss of zero if producing the same representation for any input sample but the identities for different samples are non-distinguishable. The reason behind is that they only maximize the numerator of $q_{NCE}$ in Eq. (1) instead of an estimation of $q(y|x)$. This also explains why most of the other theoretical works [2,3,4,5] also focus on contrastive learning as us.
>
> - **Nevertheless, IDAA (our method) can significantly improve the efficiency and effectiveness of those non-contrastive learning methods such as SimSiam and FixMatch, as demonstrated in our experiments** (Figure 5, Table 1, 3 and 6). This verifies our statement above, i.e., consistency-type methods are motivated by identity preserving though it does not theoretically hold in the worst case. Since IDAA provably (Theorem 1) aim at preserving the identity of data augmentations, it matches the intuition of many self-supervised learning methods beyond contrastive learning and thus can generally improve their empirical performance.
>
> ```
> (2) The results reported in the paper seem not consistent with existing papers in SSL area. For example, SimCLR and SimSiam are known to achieve over 90% accuracy in linear classification when ResNet18 backbone with 800 epochs, and the results in Table 1 seem to have significant differences.
> ```
> - **Your observed difference on the final accuracy is due to fewer training epochs (300 for CIFAR and 100 for ImageNet in our results vs. $>800$ epochs in SimSiam and SimCLR paper)** since we have limited computational resources to complete all experiments. We would like to argue that **the difference is small**, e.g., 89.84% of our reproduced SimSiam vs 91.8% reported in original SimSiam[6] on CIFAR10, especially considering our saving on the number of training epochs. Another critical reason for choosing less epochs is that one major goal of IDAA is to **improve the efficiency of self-supervised learning (SSL)** via more informative data augmentation, because **most of existing SSL methods require $10$-$100$ times epochs** than supervised learning to reach a comparable accuracy, which is costly and create tremendous carbon emission in practice.
>
> - **We use fewer epochs for a better comparison with the results reported in other data augmentation methods recently developed for SSL such as CLAE [7] and InfoMin [8] because they also use fewer epochs**. In fact, we use exactly the same epochs as CLAE [7], which is a major baseline since its augmentation is also based on adversarial perturbations. Another baseline, InfoMin [8], also runs SSL methods for only 200 epochs in their evaluation.
>
> - **In Fig. 6(c), we provided a side-by-side comparison between SimCLR and SimCLR+IDAA under different number of training epochs (varying from 100 to 1000 epochs)**. SimCLR+IDAA consistently outperform SimCLR by a large margin in all cases.

---

> > ### Author Response · Authors · 2021-11-25
> > **[Response 2/2] Thank you for the comments! But most of them are Incorrect or Inaccurate.**
> >
> >
> > ```
> > (3) In Table 1 CLAE seems to harm the performance of SimCLR and SimSiam, which is not consistent with the conclusion in their paper either.
> > ```
> >
> > - **CLAE[7] reported a much lower accuracy than the one reported in the original SimCLR[9] paper and the one we achieved on SimCLR**. This is shown in the table below, in which we provide a side-by-side comparison with the results from CLAE [7] paper. We posit the reason is that CLAE [7] did not use a decaying learning rate as instructed in SimCLR paper, which uses a decaying cosine learning rate (and we use it too!). This **can be verified by looking into the official code** of CLAE [7], i.e., line 20 of https://github.com/chihhuiho/CLAE/blob/main/SimCLR/model.py.
> >
> > |Method||CIFAR10|CIFAR100|
> > |--:|:--|:--|:--|
> > |SimCLR|(CLAE [7] results)|83.27$\pm$0.17|53.79$\pm$0.21|
> > |SimCLR+CLAE|(CLAE [7] results)|83.32$\pm$0.26|55.52$\pm$0.30|
> > |SimCLR|(Our results)|86.40$\pm$0.18|57.81$\pm$0.23|
> > |SimCLR+CLAE|(Our results)|85.25$\pm$0.07|57.69$\pm$0.25|
> >
> > - CLAE does not bring improvement to SimSiam in our experiments, though we use CLAE's official code and tune its hyper-parameters to get its best performance. This is **consistent with our observation on SimCLR**. It is also worth noting that SimSiam is not included in CLAE's paper [7] but SimSiam performs much better than those baselines in their paper.
> >
> > ```
> > (4) According to beta-VAE, the disentanglement is affected by choice of beta, and they have better disentanglement in the latent space when beta is bigger than 1. According to Eqn (5) in the paper, it also seems the identity is better preserved from wrong augmentation when beta is bigger (please correct me if I am wrong).
> > ```
> >
> > - **It is incorrect (and harmful) to always increase or use a large $\beta$** because a good augmentation requires both the identity disentanglement ($I(y;z)$ in Eq. (5)) and the VAE reconstruction ($I(x;z)$ in Eq. (5)), and $\beta$ controls their trade-off. When using a larger $\beta$, VAE reconstruction will be poorer and $I(x;z)$ will be smaller, which is detrimental for SSL, because a smaller $I(x;z)$ means poorer augmentations far away from the true data distribution. This has been discussed in line 16-17 in Sec 3.3.
> >
> > - Therefore, a correct interpretation of Eq. (8) is: to improve the lower bound, one should mainly minimize $L_{VAE}$ because minimizing $I(x;z)$ is harmful (e.g., removing useful information about $x$, producing augmentations far away from the true data distribution, etc.).
> >
> > - **The "disentanglement" in $\beta$-VAE [10] (and referred by you) is NOT the "identity-disentanglement" in our paper, so $\beta=1$ in their paper might not lead to the best identity disentanglement**. Specifically, their disentanglement is between the latent features in VAE's bottleneck layer, while our disentanglement is between the identity-preserved part and the identity-irrelevant part of data in the input space.
> >
> > ```
> > (5) A weakness here is IDAA seems to rely upon existing pre-defined augmentation combinations. The authors better tune down their claim or provide more support that IDAA can achieve on par performance even with random augmentation or without any augmentation.
> > ```
> >
> > - **We respectfully disagree with your opinion. IDAA is a complementary technique rather than an alternative to existing random augmentations**. Random augmentations encode important human prior and have been widely used. SSL methods such as SimCLR spend a great amount of computation to search for the best combination of random augmentations. So there is no reason for not using them! The challenging problem we address here is: can we further improve self-supervised learning on top of these hand-crafted "optimal" random augmentations? As demonstrated by thorough experiments in this paper, **IDAA can consistently and significantly improve SSL methods already equipped with those "optimal" random augmentations**.
> >
> > - **To directly answer your question, IDAA can still significantly improve the performance of SSL methods such as SimCLR even when no random augmentation is used**, as shown by the results on CIFAR-100 in the table below, though the performance is much poorer than that with random augmentations. This is definitely not practical but a sanity check for your question.
> >
> > ||kNN|Linear Evaluation|
> > |:--|:--|:--|
> > |SimCLR (w/o random augmentation)|21.16|34.71|
> > |SimCLR+IDAA (w/o random augmentation)|**27.67**|**42.44**|

---

> > > ### Author Response · Authors · 2021-11-25
> > > **References**
> > >
> > >
> > > [1] Kaiming He, Haoqi Fan, Yuxin Wu, Saining Xie, Ross Girshick. Momentum Contrast for Unsupervised Visual Representation Learning, CVPR 2020.
> > >
> > > [2] Sanjeev Arora, Hrishikesh Khandeparkar, Mikhail Khodak, Orestis Plevrakis, Nikunj Saunshi. A Theoretical Analysis of Contrastive Unsupervised Representation Learning, ICML 2019.
> > >
> > > [3] Colin Wei, Kendrick Shen, Yining Chen, Tengyu Ma. Theoretical Analysis of Self-Training with Deep Networks on Unlabeled Data, ICLR 2021.
> > >
> > > [4] Christopher Tosh, Akshay Krishnamurthy, Daniel Hsu. Contrastive learning, multi-view redundancy, and linear models, ICMLR 2021.
> > >
> > > [5] Jeff Z. HaoChen, Colin Wei, Adrien Gaidon, Tengyu Ma. Provable Guarantees for Self-Supervised Deep Learning with Spectral Contrastive Loss, NeurIPS 2021.
> > >
> > > [6] Xinlei Chen and Kaiming He. Exploring simple siamese representation learning. CVPR 2021.
> > >
> > > [7] Chih-Hui Ho, Nuno Vasconcelos. Contrastive Learning with Adversarial Examples, NeurIPS 2020.
> > >
> > > [8] Yonglong Tian, Chen Sun, Ben Poole, Dilip Krishnan, Cordelia Schmid, Phillip Isola. What Makes for Good Views for Contrastive Learning? NeurIPS 2020.
> > >
> > > [9] Ting Chen, Simon Kornblith, Mohammad Norouzi, Geoffrey Hinton. A Simple Framework for Contrastive Learning of Visual Representations, ICML 2020.
> > >
> > > [10] Irina Higgins, Loic Matthey, Arka Pal, Christopher Burgess, Xavier Glorot, Matthew Botvinick, Shakir Mohamed, Alexander Lerchner. beta-VAE: Learning Basic Visual Concepts with a Constrained Variational Framework, ICLR 2017.

---

> > > ### Comment · Reviewer_rByQ · 2021-11-25
> > > **Re: Response 2/2**
> > >
> > > > ### Results of CLAE and choice of $\beta$
> > >
> > > I appreciate the authors providing better reproduced CLAE results and addressing my concern in this part.
> > >
> > > I also agree that naively increasing $\beta$ could be harmful and the disentanglement in $\beta$-VAE could be different from identity-disentanglement. As $\beta$ is an important hyperparameter, and the best choice could be different on different datasets with different VAE bottleneck dimensions. It would be better the authors could provide ablation studies like Fig. 7 in different settings.
> > >
> > > > ### IDAA and pre-defined augmentation combinations
> > >
> > > Thanks again for clarifying this point. It is non-trivial to point out IDAA is a complementary technique rather than an alternative to existing pre-defined augmentations (I am not the only one who got misled that IDAA can work independently to existing augmentation methods as I also noticed some of my co-reviewers also have similar questions to this part.)
> > >
> > > I apologize that I did not see your points in the results provided above. If IDAA is a complementary technique, SimCLR and SimCLR+IDAA without augmentation do not make sense to me. Could you please provide more details on how you define the positive distribution without augmentations?
> > >
> > > I would like to provide my understanding and clarify my question regarding this part (again, please correct me if my understanding is not correct):
> > > SimCLR and existing methods define an augmentation combination, and augmentations are independently and randomly sampled from this augmentation set, and I guess this is the random augmentation you are referring to. My question is how does IDAA perform with random pre-defined augmentation combinations? Mathematically, let's say SimCLR defined an augmentation distribution $p_{SimCLR}(x)$, what if we change $p_{SimCLR}(x)$ to a random distribution $p(x)$, and what are the effects if $p_{SimCLR}(x)$ and $p(x)$ have small/large overlap?

---

> > > > ### Author Response · Authors · 2021-11-25
> > > > **Thank you for confirming that your concerns have been addressed! Here is our reply to your further concerns.**
> > > >
> > > > Thank you for reading our response! We are glad to hear that your concerns have been addressed by our response. Here is our response to your new comments:
> > > >
> > > > > ### Results of CLAE and choice of $\beta$
> > > >
> > > > - **Fig. 7 already provides a thorough sensitivity study of all the hyperparameters** including $\beta$ and it shows that our method's performance and **IDAA-induced improvement are robust to the change of these hyperparameters**. For example, in a wide range of $\beta\in[0.1,10]$, our performance's change is within only 1% and SimCLR+IDAA is always better than SimCLR in this wide range of $\beta$!
> > > >
> > > > - While we agree with you that having Fig. 7 for more datasets can further strengthen the paper, it is worth noting that **Fig. 7 requires running (14 hyperparameter choices x 2 methods x 10 random seeds) = 280 experiments for every dataset, and this means 280 x 300 epochs = 84000 epochs per dataset!** We can try our best to launch these experiments but our computational resources may not guarantee to complete them before the end date of the discussion stage.
> > > >
> > > > >### IDAA and pre-defined augmentation combinations
> > > >
> > > > - **If we understand your question correctly, it has already been answered in the original SimCLR paper** because the "random augmentations" in SimCLR are the best pre-defined augmentation operations among 49 choices, as shown in Figure 5 of their paper: https://arxiv.org/pdf/2002.05709.pdf. Specifically, if we define $p(x)$ as the distribution of augmentations randomly drawn from the 49 choices in Figure 5, $p_{SimCLR}$ is always better than $p(x)$ because it adopts the best choice (random cropping + random color distortion) among the 49 ones.
> > > >
> > > > - **In this paper, we are avoiding changing anything in SimCLR** because (1) we assume that their proposed designs are already near-optimal in its framework given its wide usage; and (2) **our focus is to develop an augmentation method on top of and complementary to existing SSL such as SimCLR**. Therefore, we did not try to change SimCLR's original design of random augmentations.
> > > >
> > > > Please let us know if the above response addresses your new concerns. Thanks!

---

> > ### Comment · Reviewer_rByQ · 2021-11-25
> > **Re: Response 1/2 (theoretical part)**
> >
> > ### **I appreciate the authors providing their responses to address my concerns. I do not think my reviews are incorrect, but the authors might have some misunderstanding of my questions. I wish the clarification below could make them more clear to the authors and hope they could be useful to further improve this paper.**
> >
> > > ### The analysis in section 3 is for InfoNCE loss, but not for more general self-supervised learning cases
> >
> > I do like your analysis in section 3 and I think they provide important results to the community of contrastive self-supervised learning. However, the analysis here is not clear to the non-contrastive self-supervised method, *i.e.*, self-supervised methods that do not use negative samples like Simsiam. For this reason, I think either way below could make this paper much stronger:
> >
> > 1) narrow your scope to contrastive methods and make comparisons with these methods using InfoNCE loss (e.g. SimCLR, MoCo) or corresponding variants (e.g. Align-Uniform loss, Hard-negative InfoNCE, etc.). I personally think this option is better as your analysis is already complete to InfoNCE and there is no need to make a connection to recent non-contrastive methods;
> >
> > 2) if you could provide additional proof for non-contrastive methods, please do it, and the comparison with SimSiam will make sense here. Otherwise, the results for SimSiam will be only an empirical study and is hard to generalize. If you wish to justify this point empirically, I think it is also necessary to justify with more baselines such as BYOL, DINO, SWAV, etc.
> >
> > Please note I am not questioning your analysis in section 3 and I think my suggestions above should be correct. After reading your responses, I still think the theorems in section 3 are insufficient to support the connection to non-contrastive methods. As you said, your theory cannot be rigorously applied to them. The motivation of identity preservation is non-trivial, but more works need to be done in either theoretical/empirical ways. That's why I personally prefer that we only discuss InfoNCE-based method here.

---

> > > ### Comment · Reviewer_rByQ · 2021-11-25
> > > **Re: Response 1/2 (experiment part)**
> > >
> > > > ### The experiment settings are not consistent with baselines
> > >
> > > Thanks for the clarification that IDAA is only pretrained with 300 epochs in your experiment setting. This part is really confusing as I cannot find the detailed experiments in the paper (although ablation studies in Fig 6 mentioned IDAA has better performance at 300 epochs, but the readers won't be sure how many epochs you are using in most of the experiments). However, I still have some confusion regarding this setting:
> > >
> > > - Could you please provide SimCLR+IDAA at 800 epoch, if possible? I am not sure of your purpose in Table 1. If you aim to show IDAA improves SimCLR's performance, it would be more convincing if SimCLR+IDAA outperforms SimCLR's best performance. If you aim to demonstrate the efficiency, I don't think presenting results at 300 epochs is sufficient to demonstrate IDAA makes contrastive learning more efficient. In fact, Fig. 6c is better to demonstrate the efficiency in the training, and Table 1 has less information here.
> > > - I don't think 89.84% vs 91.8% is a small difference and computational resources are not the bottleneck for training epochs. The experiment setting is wrongly picked here. If your target is to show IDAA improves CLAE or InfoMin, you can use their setting with fewer epochs, but I think here you are comparing with SimCLR.

---

> > > > ### Author Response · Authors · 2021-11-25
> > > > **Author Response to "Re: Response 1/2 (experiment part)"**
> > > >
> > > > Thank you for your quick respose！While we appreciate your comments, we do think you **still misunderstand the key problem we mainly study and target in the paper**. Here is our detailed response:
> > > >
> > > > - **We study instance-adaptive data augmentation for self-supervised learning (SSL), the same problem studied by CLAE and InfoMin. So our main target is to show IDAA outperforms CLAE and InfoMin** when applied to SSL methods such as SimSiam and SimCLR. Specifically, we aim at showing that the improvement IDAA can bring to SimSiam/SimCLR is larger than the improvement brought by CLAE and InfoMin.
> > > >
> > > > - **We are NOT studying a new self-supervised learning method such as DINO, SWAV, and BYOL and our goal is NOT to replace SimCLR. The main purpose for having the results of SimCLR and other SSL methods here is to compare IDAA with CLAE and InfoMin**, i.e., to show that the improvement IDAA can bring to SSL methods is larger than the improvement brought by CLAE and InfoMin. Therefore, IDAA (ours) as a data augmentation technique falls into the same category of CLAE and InfoMin, instead of the category of DINO, SWAV, BYOL, and MoCo you mentioned.
> > > >
> > > > - **Because we mainly target the problem of CLAE and InfoMin, we follow their setting of fewer epochs and this is the correct experimental setting!** Although our main results focus on fewer epochs, we did provide results under different numbers of epochs up to 1000 epochs in Fig. 6(c), which show an obvious, significant, and consistent improvement brought by applying IDAA to SimCLR under more epochs.
> > > >
> > > > - **Besides our better improvement than CLAE and InfoMin, we take a further step towards improving a broader class of SSL methods including non-contrastive learning ones**, while CLAE and InfoMin only focus on contrastive learning. One key insight here is: identity preserving is a hidden target shared by various types of SSL methods so identity-disentangled data augmentations such as IDAA can generally improve a broad class of SSL methods.

---

> > > ### Author Response · Authors · 2021-11-26
> > > **Author Response to "Re: Response 1/2 (theoretical part)"**
> > >
> > > Thank you for the quick response and valuable suggestions! **We also appreciate your positive comments about our analysis in Section 3, which is our key idea/novelty motivating the design of IDAA**. In the next version, we will follow your suggestion to **narrow our claims of main contribution to identity-disentangled augmentation for contrastive learning methods**. In the theoretical part and most technical parts, we will mainly retain the current discussion of contrastive loss and VAE only. For the empirical evaluation, we believe that our improvement achieved on SimSiam and FixMatch is non-trivial, explainable by the identity preserving idea (though not by our theory), and is still valuable and helpful to machine learning practitioners in the community. Here is our detailed reply.
> > >
> > > - We will polish the claim of our main contribution to focus on contrastive learning methods and this is already an important contribution because **(1) the theoretical analysis in Section 3 is novel, rigorous, and complete** as acknowledged by you; and **(2) the empirical evidences are also sufficient**, i.e., we applied IDAA to **THREE different contrastive learning methods** (Plain, UEL and SimCLR) on **11 datasets** (CIFAR10, CIFAR100, miniImageNet, ImageNet 64 $\times$ 64, Aircraft, DTD, Pets, Flower and CUB-200), and **IDAA consistently brings more significant improvement than other widely-used or recently-proposed data augmentation baselines** (random augmentation, CLAE and InfoMin).
> > >
> > > - Though tuning down our claim about improving general SSL methods, we believe that our current experiments of IDAA and **its non-trivial improvement to non-contrastive methods for self-supervised and semi-supervised learning (i.e., Figure 5, Table 1, 3 and 4) are valuable and helpful to machine learning practitioners in these two fields**. These results show that **in practice IDAA can substantially improves the effectivenss, efficiency and transferability of non-contrastive methods** like SimSiam and FixMatch, the two popular methods in these two fields. They empirically implies that identity preserving and disentanglement in data augmentation, which is the primary novelty of IDAA, **is simple, principal, and effective beyond contrastive learning**. We believe that the theoretical gap of IDAA on non-contrastive learning can be fixed in the near future by addressing the collapse problem of non-contrastive learning.

---

> ### Author Response · Authors · 2021-11-27
> **We added experiments you requested: Ablation study of different values for $\beta$ on three datasets**
>
> We have followed your suggestions and report **an thorough ablation study regarding $\beta$ on all the Three datasets**. Specifically, we tried four values of $\beta$, i.e., $0, 0.1, 1, 10$, in SimCLR+IDAA on three datasets and report the results in the table below. The performance of our method is robust to the change of $\beta$ and keeps surpassing all the baselines. Here are the conclusions from the results.
>
>
> - The best value of $\beta$ for both CIFAR10 and CIFAR100 is 0.1, while that for miniImageNet is 10. This is because the images in miniImageNet are
> of higher resolution and contain richer identity-information than CIFAR, hence **a larger $\beta$ is needed to enforce a stronger identity-disentanglement**.
>
> - **The performance of IDAA is robust to the choice of $\beta$**, e.g., the maximal difference on accuracy is merely $<0.5$% between two choices of $\beta$ in the miniImageNet experiments. Therefore, SimCLR+IDAA consistently outperforms SimCLR for all the evaluated values of $\beta$.
>
> - In the main paper, we did not tune the value of $\beta$ for each dataset separately but instead use the same $\beta=0.1$ for all the datasets. As shown in the table, we can **achieve better results than the previous results in our paper if we carefully tune $\beta$ for every dataset**.
>
> |$\beta$|CIFAR10|CIFAR100|miniImageNet|
> |--:|:--|:--|:--|
> |0|86.79|59.18|48.19|
> |0.1|**88.07**|**60.90**|48.23|
> |1|87.75|60.47|48.44|
> |10|87.64|59.81|**48.51**|
>
> We believe that we addressed all your concerns and completed all the experiments you requested with positive results. Please let us know if you still have more concerns. Thanks!

---

> ### Author Response · Authors · 2021-11-30
> **We addressed all your concerns with new experiments. Would you mind confirming if you have further concerns and reconsidering your rating? Thanks!**
>
> Dear Reviewer rByQ:
>
> We appreciate your comments and time! We addressed all your concerns with detailed responses below. Would you mind checking them and confirm if you have further questions? Here is a short summary:
>
> - **All the experiments in our paper are fair and standard**: experiments in Table 1-2 and Table 4-8 exactly follow CLAE, i.e., train a ResNet-18 for 300 (100) epochs for CIFAR (ImageNet); experiments in Table 3 exactly follow FixMatch, i.e., train a Wide-ResNet-28-8 with 400, 2500 and 10000 labeled data.
>
> - **Performance on more training epochs (up to 1000 epochs)** has been thoroughly evaluated in Figure 6 (c) of Sec 4.4 for the ablation study.
>
> - **We clarified the scope of our model**: the theoretical analysis is novel and holds for most contrastive learning methods, while the empirical improvements and the intuition of identity-preservation hold true for both contrastive and non-contrastive methods.
>
> - Studying augmentations **on top of existing augmentations** is a **common setting adobted by ALL augmentation methods for SSL**, i.e., CLAE, AdCo and InfoMin. They did **NOT** remove the augmentation of SimCLR/CMC when applying their augmentations and **we followed this standard setting for a fair comparison**.
>
> - We provided **a thorough ablation study for different values of $\beta$ on three different datasets**.
>
> Would you mind letting us know if you have any further concerns? Thanks a lot!
>
> Best Regards,
>
> Authors

---

> > ### Comment · Reviewer_rByQ · 2021-12-01
> > **Response to authors**
> >
> > I appreciate the efforts of the authors in addressing my comments. Tuning down the claim and providing more guidance in the choice of $\beta$ do make the presentation quality better. However, my main concerns in the main experiments are still not addressed:
> >
> > Results in Table 1 are not sufficient to show the benefits of IDAA. Even you are comparing with CLAE, it is shown in Table 1 that CLAE does not outperform SimCLR.  It is not that important to show IDAA is better than CLAE. Instead, showing IDAA can further improve SimCLR is more important. Thus not only I but also my co-reviewers suggest making your experiment setting consistent with SimCLR and SimSiam, and if possible, please show us results with ImageNet or ImageNet-100. The tinyImageNet results are not as good as training a classifier from scratch, thus the significance of these experiments is limited.
> >
> > I still think this paper will have insight into the community, but based on the current presentation quality, there are so many things unclear to the readers. Therefore, I would like to keep my original rating.

---

> > > ### Author Response · Authors · 2021-12-02
> > > **We are studying DATA AUGMENTATION for SSL, Not a new SSL method! We provided ImageNet experiments in various settings!**
> > >
> > >
> > > Thanks for your reply! **We are glad to hear that your initial concerns have been addressed**. However, we respectfully **disagree** with you on your further concerns about our main experimental setting: **this paper is studying a DATA AUGMENTATION method for SSL, not a new SSL method**, so we followed the setting of the most related and recent paper that studies DATA AUGMENTATION for SSL. Although our computational resource cannot support us to run all the hundreds of experiments on the full-resolution ImageNet for 1000 epochs in such a short window of time, **We did provide sufficient ImageNet experiments and comparisons in various settings, using the upper limit of our computational resource**. Our ImageNet experiments cover full resolution case, full dataset (1000 classes) case, cases of training ResNet-50 etc.
> > >
> > > **Machine learning (and ICLR) community should be more inclusive of researchers who do not have the same computation resources as SimCLR team but DO provide novel theoretical insights and diverse empirical evidence. Not every one is rich in computational credits. It is unfair to always use one single experimental setting as the unique golden evaluation criterion**: there are plenty of applications in this world caring about training models under limited computational resources, e.g., fewer epochs, lower-resolution data, smaller models, fewer classes, etc.
> > >
> > > - You said "It is not that important to show IDAA is better than CLAE"--- **This is incorrect**!  **CLAE does improve 2 out of the 4 SSL methods in Table 1**, i.e., Plain and UEL (note our method IDAA improves ALL the four SSL methods!). Moreover, **CLAE consistently improves the transfer learning performance** of SimCLR (6 out of 8 datasets, in Table 2) and SimSiam (8 out of 8 datasets, in Table 4) on most datasets. In addition, **CLAE also improves the semi-supervised learning performance of FixMatch** in Table 3. Since SimCLR is not the only SSL method and SSL is not the only problem that data augmentation methods can be applied to, CLAE is still an important baseline.
> > >
> > > - We chose CLAE as our main baseline and its setting as our main experimental setting because it is the most related paper studying the same problem as our paper, i.e., adversarial data augmentation for SSL.
> > >
> > > - **Our data augmentation method (IDAA) does improve the original SimCLR under SimCLR's setting in our reported experiments!** Figure 6 (a),(c) compare SimCLR with SimCLR+IDAA on training ResNet-50 and ResNet-101 up to 1000 epochs, which is **exactly the same setting as SimCLR. Would you mind reading Sec 4.4?**
> > >
> > > - **Unlike many pure empirical works, we developed new theories for data augmentation of SSL and it further motivated our practical algorithm for both self-supervised and semi-supervised learning.** And we did provide sufficient ImageNet experiments in various diverse settings, **using the upper limit of our computational resource!** Here is a list of all the ImageNet experiments reported in this paper and responses to reviewers. In all the experiments, IDAA (our data augmentation method) substantially improves the SSL methods on the standard evaluation metrics.
> > > ###### In Table 6: kNN and linear evaluation of ResNet-18 pretrained by SimCLR vs. "SimCLR+IDAA" on **ImageNet 64x64 (all 1000 classes)** for 100 epochs;
> > > ##### In response for Reviewer dMSm: kNN and linear evaluation of ResNet-50 pretrained with InfoMin vs. IDAA augmentations **ImageNet 64x64 (all 1000 classes)** for 100 epochs;
> > > ##### In Table 2: transfer learning performance of ResNet-18 pretrained by SimCLR vs. "SimCLR+IDAA" on **ImageNet100 (full resolution)** for 100 epochs;
> > > ##### In Table 4: transfer learning performance of ResNet-18 pretrained by SimSiam vs. "SimSiam+IDAA" on **ImageNet100 (full resolution)** for 100 epochs;
> > > ##### In the table below: kNN and linear evaluation of ResNet-18 pretrained by SimCLR vs.  "SimCLR+IDAA" on **ImageNet100 (full resolution)** for 300 epochs.
> > >
> > > |ImageNet100|kNN|Linear Evaluation|
> > > |:--|--:|--:|
> > > |SimCLR|57.38|72.78|
> > > |SimCLR+IDDA|**61.90**|**74.03**|
> > >
> > > In all experiments under diverse settings of ImageNet, our data augmentation method (IDAA) consistently improves SSL methods (SimCLR and SimSiam) by a large margin.

---

### Author Response · Authors · 2021-11-22
**Summary of Changes in Revision**

We would like to thank all reviewers for their efforts in the reviewing process. We uploaded a revision of the paper, in which we add six groups of new experiments and address all the concerns of the reviewers. You can find them highlighted in red color. Here is a summary of the major changes:

- [**New experiments**] We add new experimental comparisons with two additional baselines published recently in Appendix A.7.

- [**New experiments**] We add new experiments to compare the transfer learning performance of different augmentation methods applied with another self-supervised learning method ``SimSiam'' in Appendix A.4.

- [**New experiments**] We add new experiments of "FixMatch+CLAE" on CIFAR100 under different amounts of labeled data in Table 3 for a complete comparison between CLAE and IDAA.

- [**New experiments**] We add experiments regarding two larger batch sizes of 1024 and 2048 in Figure 6(a) to evaluate the performance of different methods when using large batch sizes.

- [**New experiments**] We add new experiments of an additional baseline without identity decomposition, i.e., with augmentation of $x'=G(z')$, in Appendix A.5, as an ablation study.

- [**New experiments**] We add new experiments on lower resolution version (64×64) of ImageNet-1k in Appendix A.6.

- We add more explanation to Theorem 1 in Appendix A.2 to make the proof more rigorous.

- We add a conclusion section to the main paper and move some experimental details to the Appendix.

- We make all the plots in the paper larger and more legible.

- We remove the augmentations produced by CLAE and IDAA in Figure 4 since it is hard to notice their difference visually. We can instead compare their generated perturbations, which are very different from each other.

- We fix the typo in the Introduction: "ullustrated in Fig.2".

---

### Decision · Program_Chairs · 2022-01-20

**Decision:**

Reject

**Comment:**

To improve the data augmentation for an existing self-supervised learning framework, the paper presents identity-disentangled adversarial augmentation (IDAA) that utilizes a pretrained VAE and adversarial perturbation in the VAE latent space to generate identity-preserving hard negative/positive samples. The proposed method has been clearly described and is of interest to the community. A wide variety of experiments have been done to illustrate the effectiveness of IDAA. Given the concerns on the validity and generalizability of the experimental results presented in the paper, the AC does not consider the paper of its current form to be ready for publication, as discussed below.

The AC appreciates the amount of effort that the authors have spent responding to the reviewers. However, thees very detailed rebuttals do not appear to be that effective in directly answering several key concerns of the reviewers.

1. While many experiments have been added to improve the paper, most of the settings are still considered by the reviewers to be unconvincing due to the use of a small number of epochs, unusual network backbones, and/or non-standard datasets (e.g., downsampled 64*64 ImageNet with about 30% accuracy).

2. The comparison to the literature is considered to be insufficient. While the basic idea is still how to generate effective views or hard negative/positive views, the paper chooses to narrowly focus its comparison with CLAE that relies on adversarial perturbation in the pixel space, ignoring a careful comparison with many existing methods that boost the performance by mining hard negatives/positives and/or using a memory bank to accommodate a large number of negatives.

3. Related to point 2, a main justification of IDAA is that it outperforms CLAE, which, however, is shown by the authors to be a weak baseline that barely improves an existing self-supervised learning (SSL) framework in a variety of different settings. For example, the results in Table 1 show CLAE hurts SimCLR's performance. The weak performance of CLAE makes it even more important to include stronger baselines.

4. A reviewer also pointed out that the use of a pretrained VAE introduces additional model parameters and increases the computation cost, and hence a careful discussion on how to ensure a fair comparison with the baselines is desired.

5. The presentation could be somewhat misleading in giving the impression that IDAA is a stand-alone domain-agnostic data augmentation technique. While it is totally fine that IDAA is a complementary data augmentation technique, the authors need to carefully revise the presentation to minimize the risk of giving the reader a wrong impression.

6. The authors dismissed some of the negative reviews by arguing them to be “mostly incorrect, wrong, makes no sense”. These strong arguments, if not supported with clear evidence asked by the reviewers, may convince neither the reviewers nor the AC and lead to unnecessarily tedious lengthy discussions. From the AC's reading of the paper and reviews, it appears that there are several cases where a reviewer was asking the answer to Question A, but the authors were providing an answer to Question B and pushing the reviewer to accept that as a satisfactory answer to Question A. For example, while many experiments have been done on several different datasets, there is a legitimate concern on why these computing resource has not been spent on getting results on ImageNet with ResNet 50. When the experiments are only being done on ImageNet-100 or the downsampled full ImageNet with a small network, there could be clear concerns on whether the observed performance gains are only applicable to relatively small/low-resolution datasets or small networks. For example, the VAE and/or the adversarial attack may not work that well on large-scale/high-resolution images and hence IDAA may break down when scaling it to a larger dataset/model. That type of concern can only be addressed if the authors have taken the time to follow a standard setting, such as ResNet50 on ImageNet, and make comparisons with a wide variety of baselines whose results on these standard settings are readily available.